# A network medicine approach to investigation and population-based validation of disease manifestations and drug repurposing for COVID-19

Yadi Zhou[1☯], Yuan Hou[1☯], Jiayu Shen[1], Reena Mehra[2,3], Asha Kallianpur[1,2], Daniel A. Culver[4,5], Michaela U. Gack[6], Samar Farha[4,5], Joe Zein[2,4], Suzy Comhair[2,4], Claudio Fiocchi[2,4], Thaddeus Stappenbeck[2,4], Timothy Chan[2,4], Charis Eng[1,2,4,7,8], Jae U. Jung[2,4], Lara Jehi[2,3], Serpil Erzurum[2,4], Feixiong Cheng[1,2,8]*

1 Genomic Medicine Institute, Lerner Research Institute, Cleveland Clinic, Cleveland, Ohio, United States of America, 2 Department of Molecular Medicine, Cleveland Clinic Lerner College of Medicine, Case Western Reserve University, Cleveland, Ohio, United States of America, 3 Neurological Institute, Cleveland Clinic, Cleveland, Ohio, United States of America, 4 Lerner Research Institute, Cleveland Clinic, Cleveland, Ohio, United States of America, 5 Department of Pulmonary Medicine, Respiratory Institute, Cleveland Clinic, Cleveland, Ohio, United States of America, 6 Florida Research and Innovation Center, Cleveland Clinic, Port Saint Lucie, Florida, United States of America, 7 Department of Genetics and Genome Sciences, Case Western Reserve University School of Medicine, Cleveland, Ohio, United States of America, 8 Case Comprehensive Cancer Center, Case Western Reserve University School of Medicine, Cleveland, Ohio, United States of America

☯ These authors contributed equally to this work.
* chengf@ccf.org

**Data Availability Statement:** All data sets used in this study and their sources for downloading can be found in S1 Table. The bulk and single-cell RNA-

## Abstract

The global coronavirus disease 2019 (COVID-19) pandemic, caused by severe acute respiratory syndrome coronavirus 2 (SARS-CoV-2), has led to unprecedented social and economic consequences. The risk of morbidity and mortality due to COVID-19 increases dramatically in the presence of coexisting medical conditions, while the underlying mechanisms remain unclear. Furthermore, there are no approved therapies for COVID-19. This study aims to identify SARS-CoV-2 pathogenesis, disease manifestations, and COVID-19 therapies using network medicine methodologies along with clinical and multi-omics observations. We incorporate SARS-CoV-2 virus–host protein–protein interactions, transcriptomics, and proteomics into the human interactome. Network proximity measurement revealed underlying pathogenesis for broad COVID-19-associated disease manifestations. Analyses of single-cell RNA sequencing data show that co-expression of *ACE2* and *TMPRSS2* is elevated in absorptive enterocytes from the inflamed ileal tissues of Crohn disease patients compared to uninflamed tissues, revealing shared pathobiology between COVID-19 and inflammatory bowel disease. Integrative analyses of metabolomics and transcriptomics (bulk and single-cell) data from asthma patients indicate that COVID-19 shares an intermediate inflammatory molecular profile with asthma (including *IRAK3* and *ADRB2*). To prioritize potential treatments, we combined network-based prediction and a propensity score (PS) matching observational study of 26,779 individuals from a COVID-19 registry. We identified that melatonin usage (odds ratio [OR] = 0.72, 95% CI 0.56–0.91) is

Seq data used in this study were downloaded from the NCBI GEO database with accession numbers GSE63142, GSE130499, and GSE134809. The lung and human bronchial epithelial single-cell data were downloaded from https://data.mendeley.com/datasets/7r2cwbw44m/1. Source code, the human protein-protein interactome, and drug-target network can be downloaded from https://github.com/ChengF-Lab/COVID-19_Map. All other relevant data are within the paper and its Supporting Information files.

**Funding:** This work was supported by the National Institute of Aging (R01AG066707 and 3R01AG066707-01S1) and the National Heart, Lung, and Blood Institute (R00HL138272) to F.C. This work has been also supported in part by the VeloSano Pilot Program (Cleveland Clinic Taussig Cancer Institute) to F.C. The funders had no role in study design, data collection and analysis, decision to publish, or preparation of the manuscript.

**Competing interests:** The authors have declared that no competing interests exist.

**Abbreviations:** ACEI, angiotensin-converting enzyme inhibitor; AP-MS, affinity purification–mass spectrometry; ARB, angiotensin II receptor blocker; AT2, alveolar type II; Co-IP+LC/MS, co-immunoprecipitation and liquid chromatography–mass spectrometry; COPD, chronic obstructive pulmonary disease; COVID-19, coronavirus disease 2019; DEG, differentially expressed gene; DEP, differentially expressed protein; $dN/dS$ ratio, nonsynonymous to synonymous substitution rate ratio; ES, enrichment score; FDA, US Food and Drug Administration; FDR, false discovery rate; GSEA, gene set enrichment analysis; HCoV, human coronavirus; HGMD, Human Gene Mutation Database; IBD, inflammatory bowel disease; KEGG, Kyoto Encyclopedia of Genes and Genomes; OR, odds ratio; PPI, protein–protein interaction; PS, propensity score; RNA-Seq, RNA sequencing; RNAi, RNA interference; SARS-CoV-2, severe acute respiratory syndrome coronavirus 2; viORF, viral open reading frame.

significantly associated with a 28% reduced likelihood of a positive laboratory test result for SARS-CoV-2 confirmed by reverse transcription–polymerase chain reaction assay. Using a PS matching user active comparator design, we determined that melatonin usage was associated with a reduced likelihood of SARS-CoV-2 positive test result compared to use of angiotensin II receptor blockers (OR = 0.70, 95% CI 0.54–0.92) or angiotensin-converting enzyme inhibitors (OR = 0.69, 95% CI 0.52–0.90). Importantly, melatonin usage (OR = 0.48, 95% CI 0.31–0.75) is associated with a 52% reduced likelihood of a positive laboratory test result for SARS-CoV-2 in African Americans after adjusting for age, sex, race, smoking history, and various disease comorbidities using PS matching. In summary, this study presents an integrative network medicine platform for predicting disease manifestations associated with COVID-19 and identifying melatonin for potential prevention and treatment of COVID-19.

## Introduction

The ongoing global coronavirus disease 2019 (COVID-19) pandemic has led to 38 million confirmed cases and 1 million deaths worldwide as of October 14, 2020. The United States alone has recorded nearly 8 million confirmed cases, with a death toll of more than 216,000 [1]. Several retrospective studies have reported the clinical characteristics of individuals with symptomatic COVID-19, and an emerging theme has been the significantly higher risk of morbidity and mortality among individuals with 1 or more comorbid health conditions, such as hypertension, asthma, diabetes mellitus, cardiovascular or cerebrovascular disease, chronic kidney disease, and malignancy [2–7]. However, these retrospective clinical studies are limited by small sample sizes and unmeasured confounding factors, leaving the underlying patho-mechanisms largely unknown. More specifically, it is unclear whether associations of disease manifestations and COVID-19 severity are merely a reflection of poorer health in general, or a clue to shared pathobiological mechanisms.

Severe acute respiratory syndrome coronavirus 2 (SARS-CoV-2), the virus that causes COVID-19, is an enveloped virus that carries a single-stranded positive-sense RNA genome [7,8]. SARS-CoV-2 is a newly discovered member of the coronavirus (CoV) family [9]. SARS-CoV-2 enters host cells via binding of its spike protein to the angiotensin converting enzyme 2 (ACE2) receptor on the surfaces of many cell types [10]. This binding is primed by transmembrane protease serine 2 (TMPRSS2) [10] and the host cell protease furin [11] (Fig 1A). Studies have shown that *ACE2* and *TMPRSS2* are highly co-expressed in alveolar type II (AT2) epithelial cells in the lung [12], nasal mucosa [13], bronchial secretory cells [14], and absorptive enterocytes in the ileum [15]. Yet, much remains to be learned about how these critical human proteins involved in the infection and replication of SARS-CoV-2 are associated with various disease comorbidities and complications. Systematic identification of the host factors involved in the protein–protein interactions (PPIs) of SARS-CoV-2 and the human host will facilitate identification of drug targets and advance understanding of the complications and comorbidities resulting from COVID-19 [16–20]. Studies using transcriptomics [21], proteomics [22], and interactomics (PPIs) methods [8] have contributed to a better understanding of the SARS-CoV-2–host interactome, which has enabled the investigation of the complications and comorbidities of SARS-CoV-2 and a facilitated search for effective treatment (Fig 1B).

Major efforts are underway to develop safe and effective drugs to treat COVID-19: Preventive and therapeutic strategies currently being explored include vaccination, SARS-CoV-

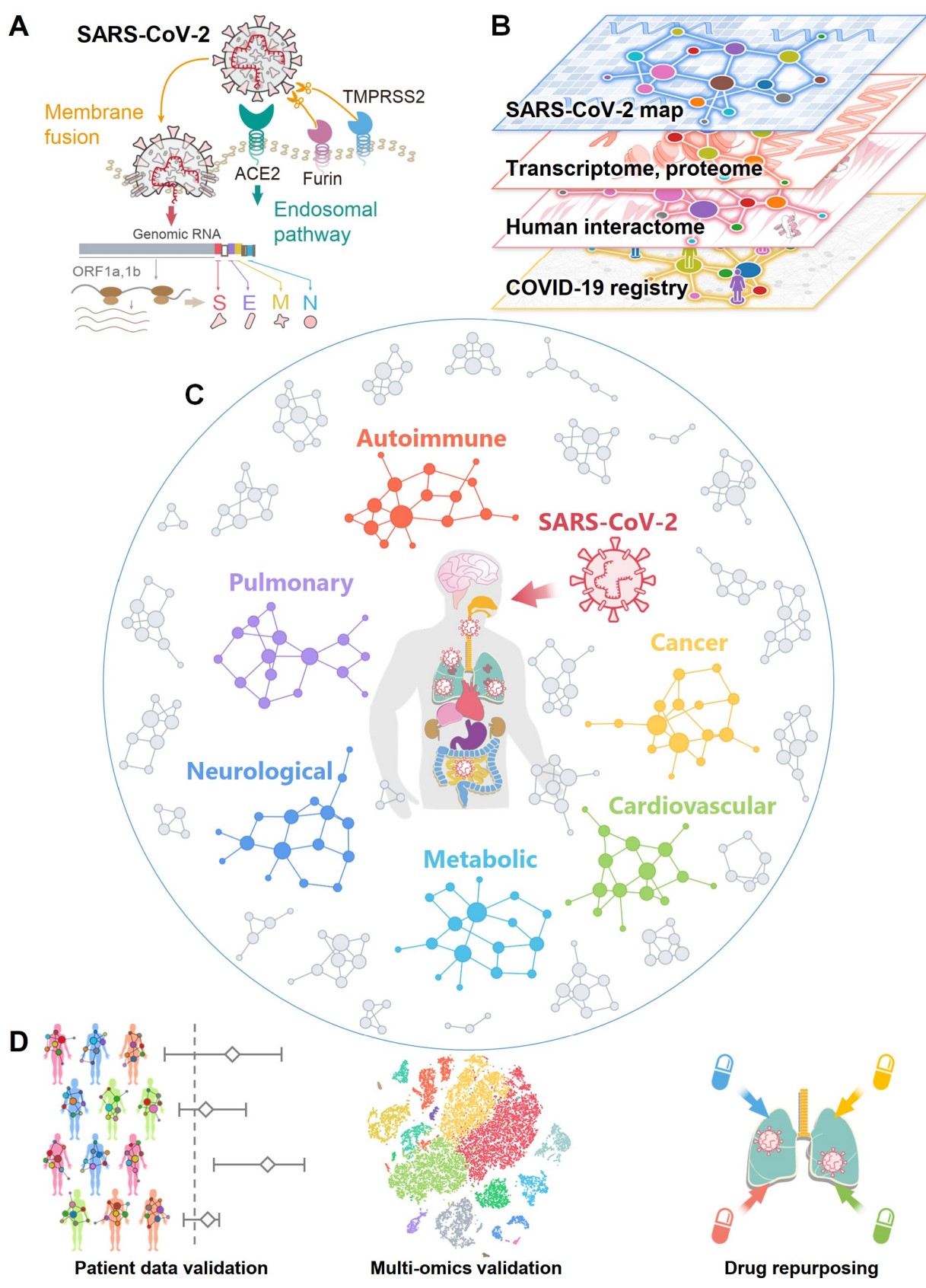

**Fig 1. Overall workflow of this study.** (A) A diagram illustrating the basic pathogenesis of SARS-CoV-2. (B) A diagram illustrating how to build a global interactome map for SARS-CoV-2. We compiled the SARS-CoV-2 human target gene/protein sets from multi-omics data from the transcriptome, proteome, and human interactome, and validated network-based findings using patient data from a COVID-19 registry. (C) A diagram illustrating network-based measurement of disease manifestations associated with COVID-19. We systematically evaluated the network proximities of the SARS-CoV-2 human target genes/proteins with 64 diseases across 6 main categories: autoimmune, cancer, cardiovascular, metabolic, neurological, and pulmonary. (D) A workflow illustrating validation of network-based findings. We performed single-cell analyses to further investigate the underlying mechanisms of COVID-19 with asthma and inflammatory bowel disease. We prioritized nearly 3,000 US Food and Drug Administration–approved/investigational drugs for their potential anti-SARS-CoV-2 effects from network-based findings and validated drug–COVID-19 outcomes using an institutional review board–approved COVID-19 patient registry.

2-specific antibodies, novel nucleoside analogs such as remdesivir, and repurposed drugs [23,24]. Remdesivir, an agent originally developed for treatment of Ebola virus, was reported to shorten the time to recovery in adults who were hospitalized with COVID-19 [25]; yet, a 10-day course of remdesivir did not show a statistically significant difference in clinical status compared with standard care for patients with moderate COVID-19 [26]. Dexamethasone, an FDA-approved glucocorticoid receptor (GR) agonist, has been shown to reduce mortality by one-third in hospitalized COVID-19 patients requiring ventilation and by one-fifth in individuals requiring oxygen [27]; yet, dexamethasone did not reduce death in COVID-19 patients not receiving respiratory support [27]. Many existing drugs are currently being or have been tested in clinical trials, such as the antimalarial drug hydroxychloroquine and protease inhibitor combination lopinavir/ritonavir; results from these trials have not yet shown significant clinical benefits for COVID-19 patients [28,29]. We recently evaluated nearly 3,000 FDA-approved/investigational drugs using a network-based method and prioritized 16 drug candidates and 3 drug combinations for COVID-19 [30]. Yet, the answer to the key question of why an approved drug originally documented for other diseases might be beneficial for COVID-19 remains unclear. One possible explanation is that COVID-19 shares common disease pathobiology or functional pathways elucidated by the human PPIs [30–32]. Systematic identification of common disease pathobiological pathways shared by COVID-19 and other diseases would offer novel targets and therapies for COVID-19.

In this study, we present an integrative network medicine platform that quantifies the association of COVID-19 with other diseases across 6 categories, including autoimmune, malignant cancer, cardiovascular, metabolic, neurological, and pulmonary (Fig 1C). The rationale for these analyses rests on the notions that (1) the proteins that functionally associate with a disease (such as COVID-19) are localized in the corresponding subnetwork within the comprehensive human PPI network [31–34] and (2) proteins that are associated with a specific disease may be directly targeted by the virus or are in the close vicinity of the target host proteins. We first performed network analysis followed by single-cell RNA sequencing (RNA-Seq) data analysis to identify the underlying pathobiological relationships between COVID-19 and its associated comorbidities. Additionally, we use our network medicine findings and patient data from a large COVID-19 patient registry database to identify and prioritize existing FDA-approved drugs as potential COVID-19 drug candidates (Fig 1D).

## Results

### A global map of the SARS-CoV-2 virus–host interactome

We assembled 4 host gene/protein sets for SARS-CoV-2 (S1 and S2 Tables): (1) SARS2-DEG, representing the differentially expressed genes (DEGs) from the transcriptomic data of SARS-CoV-2-infected primary human bronchial epithelial cells; (2) SARS2-DEP, representing the differentially expressed proteins (DEPs) from the proteomic data of SARS-CoV-2-infected human Caco-2 cells; (3) HCoV-PPI, representing the literature-based virus–host proteins

across multiple human coronaviruses (HCoVs), including SARS-CoV-1 (from the 2002–2003 pandemic) and MERS-CoV; and (4) SARS2-PPI (SARS-CoV-2-specific virus–host PPIs). Since HCoV-PPI and SARS2-PPI both involve physical virus–host PPIs, we further combined them as the fifth dataset, PanCoV-PPI.

We first performed functional enrichment analyses for the 5 different gene/protein datasets. We found that these datasets share several common pathways and ontology terms (Fig 2A; S1 Data), such as phagosome, measles, apoptosis, NF-κB signaling pathway, neutrophil-related immunity, apoptotic processes, virus transport, viral genome replication, and response to interferon, yet they differ considerably in terms of their most significantly enriched pathways (S1–S5 Figs). This is especially noticeable for SARS2-DEP and SARS2-PPI. While SARS2-DEG (S1 Fig) and HCoV-PPI (S3 Fig) show more enrichment in immune responses and viral pathways, SARS2-DEP (S2 Fig) is more related to various cellular metabolic pathways, and SARS2-PPI (S4 Fig) is more enriched in DNA replication, RNA transcription, and protein translation. These observations suggest that these different SARS-CoV-2 viral–host gene/protein sets capture complementary aspects of the biological and cellular states of the viral life cycle and host immunity. Therefore, building a global virus–host map (including interactome, transcriptome, and proteome) that incorporates data from transcriptomics, proteomics, and physical virus–host PPIs are essential for a better understanding of the pathogenesis of COVID-19. This global virus–host map for SARS-CoV-2 can offer a more complete picture of the interconnected functional pathways involved in viral pathogenesis, thereby facilitating the discovery of therapeutic targets.

## Network and biological characteristics of virus–host interactome for SARS-CoV-2

In addition to identifying the functions that these viral gene/protein sets represent, we next characterized the network patterns (node degree in the human PPI network) and bioinformatics features of these SARS-CoV-2 datasets, including $dN/dS$ ratio, evolutionary rate ratio, and lung expression specificity (Figs 2B–2G and S6). To find common as well as unique network and bioinformatic characteristics of SARS-CoV-2, we further compiled 4 additional virus–host gene/protein networks, identified by different methods, for comparison (S3 Table): (1) 900 virus–host interactions connecting 10 other viruses and 712 host genes identified by gene-trap insertional mutagenesis, (2) 2,855 known virus–host interactions connecting 2,443 host genes and 55 pathogens identified from RNA interference (RNAi), (3) 579 host proteins mediating translation of 70 innate immune-modulating viral open reading frames (viORFs), and (4) 1,292 host genes mediating influenza–host interactions identified by co-immunoprecipitation and liquid chromatography-mass spectrometry (Co-IP+LC/MS). These virus–host gene/protein networks have been well characterized [35–37] and offer high-quality datasets (S3 Table) for comparisons. We found that host proteins in PanCoV-PPI (Fig 2B) and 4 other datasets (SARS2-DEG, SARS2-DEP, HCoV-PPI, and SARS2-PPI) (S6 Fig) were more likely to be highly connected in the human PPI network. Several hub genes, such as *JUN*, *XPO1*, *MOV10*, *NPM1*, *VCP*, and *HNRNPA1*, have the highest degree (connectivity) in the PanCoV-PPI network (S2 Table). PanCoV-PPI has a comparable degree distribution with host genes/proteins identified by viORFs and Co-IP+LC/MS, although marginally higher than that identified by RNAi and gene-trap insertional mutagenesis assay (Fig 2F).

Expression patterns of genes in a specific disease-related tissue play a crucial role for elucidation of disease pathogenesis and drug discovery [31,32]. Given the major impact of SARS-CoV-2 on pulmonary function and lung injury [2], we inspected the lung-specific expression of genes in PanCoV-PPI using a *Z* score measure (see Materials and Methods—Tissue

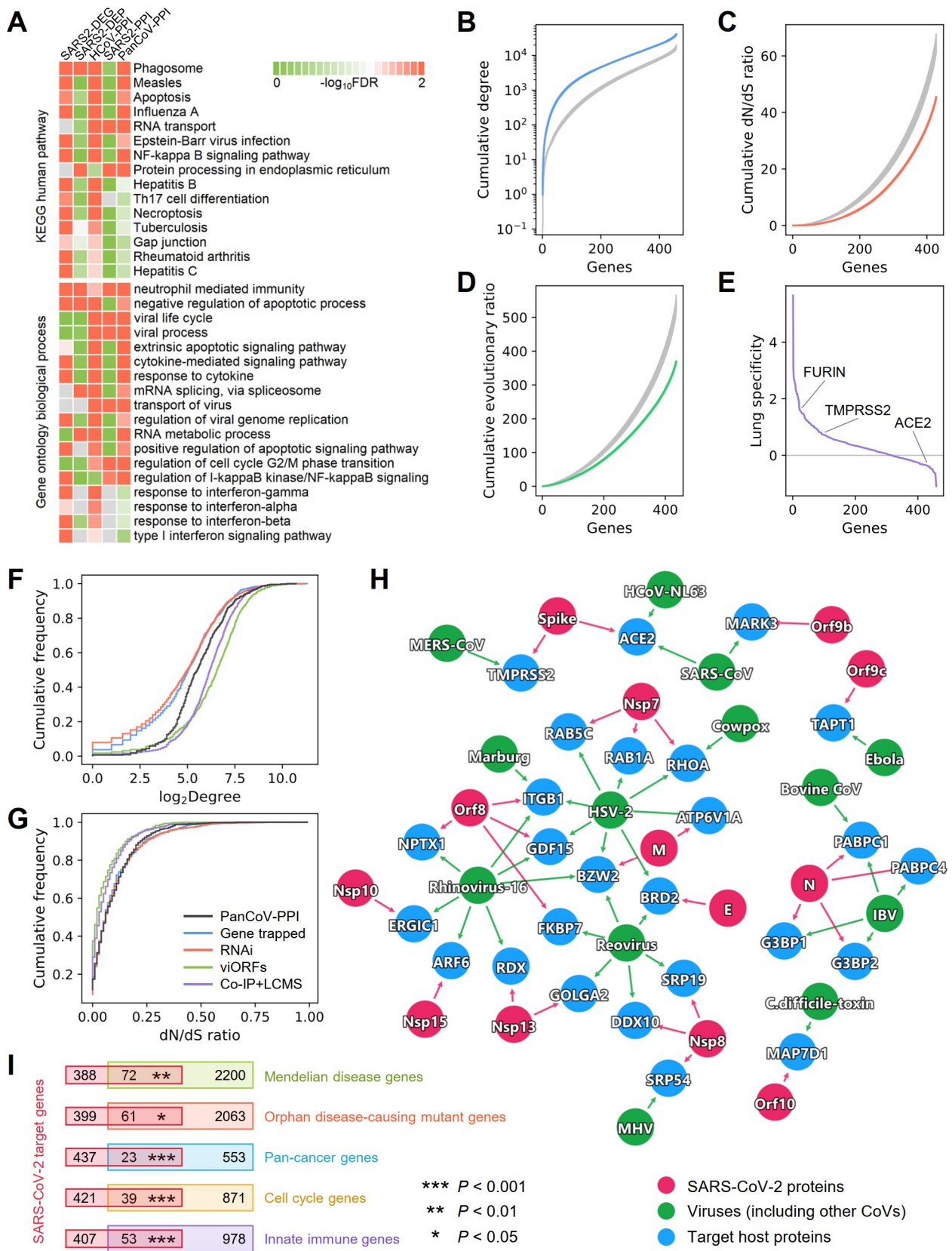

**Fig 2. Network and biological characteristics of the SARS-CoV-2 interactome map.** (A) Pathway and Gene Ontology (biological process) enrichment analysis results of the SARS-CoV-2 host genes/proteins across 5 different datasets. We assembled 5 gene/protein datasets from SARS-CoV-2 host protein–protein interactions, transcriptomics, and proteomics (S2 Table). (B–D) Network and biological characteristics of the SARS-CoV-2 host genes/proteins. The proteins in PanCoV-PPI have higher node degrees (B), lower $dN/dS$ ratios (C), and lower evolutionary ratios (D) compared to randomly selected proteins (grey, mean ± standard deviation of 100 repeats). (E) Among the 460 proteins in PanCoV-PPI, 450 (98%) are expressed in lungs, and 317 (69%) have lung-specific expression ($Z > 0$). (F and G) The distribution of the node degrees in the human interactome and $dN/dS$ ratios of PanCoV-PPI and 4 published virus-related host protein sets (S3 Table). (H) The shared target human proteins (blue) of SARS-CoV-2 (red) and other viruses (green). (I) SARS-CoV-2 target proteins overlap significantly with disease-associated genes (Mendelian disease and orphan disease), cancer genes, cell cycle genes, and innate immune genes. The data underlying this figure can be found in S1 Data. Co-IP+LCMS, co-immunoprecipitation and liquid chromatography–mass spectrometry; CoV, coronavirus; $dN/dS$ ratio, nonsynonymous to synonymous substitution rate ratio; FDR, false discovery rate; KEGG, Kyoto Encyclopedia of Genes and Genomes; RNAi, RNA interference; viORF, viral open reading frame.

specificity analysis) compared to expression in other tissues from the GTEx database [38]. We found that most host genes for SARS-CoV-2 have high expression in lung (Fig 2E; S2 Table) compared to other tissues; yet, *ACE2* has a low expression in lung compared to other tissues. A recent study showed that *ACE2* was primarily expressed in the epithelial cells in lungs, and only 3.8% of AT2 pneumocytes expressed both *ACE2* and *TMPRSS2*, but *ACE2* is significantly upregulated in smokers and 24–48 hours following SARS-CoV-2 infection [12,39]. Another study also showed that despite relatively low expression of *ACE2* in the lung, *ACE2* was expressed in multiple epithelial cell types along the airway [13]. Therefore, it is important to understand the cell-type-specific SARS-CoV-2 pathogenesis at the single-cell level.

To inspect the evolutionary factors underlying the SARS-CoV-2–human PPIs, we investigated the selective pressure and evolutionary rates quantified by the $dN/dS$ ratio using human–mouse orthologous gene pairs. We found that PanCoV-PPI shows stronger purifying selection (lower $dN/dS$ ratio [Fig 2C] and evolutionary rate ratio [Fig 2D]) compared to the same number of random genes. PanCoV-PPI is also comparable to 4 other virus–host genes/protein datasets identified by different assays (Fig 2F and 2G) in terms of node degrees and $dN/dS$ ratios. Altogether, these observations suggest that the virus–host PPIs assembled in this study offer a high-quality interactome map for SARS-CoV-2 for identifying pathogenesis and potential treatments for COVID-19.

To inspect shared viral pathways across different viruses and SARS-CoV-2, we further performed network overlap analysis of PanCoV-PPI with 712 host genes across 10 types of other viruses identified by gene-trap insertional mutagenesis assays [35]. We found a significant overlap of the SARS-CoV-2 host proteins with non-coronaviruses ($P < 0.002$, Fisher's exact test; Fig 2H). For example, BRD2, a transcriptional regulator that belongs to the Bromodomain and Extra-Terminal motif family, is connected to SARS-CoV-2 and 2 other viruses, herpes simplex virus 2 (HSV-2) and reovirus. *RHOA*, encoding a small GTPase protein in the Rho family of GTPases, is connected to SARS-CoV-2, cowpox, and HSV-2 as well. RHOA has been reported to be involved in multiple human diseases, including cardiovascular disease [40] and cancer [41]. These observations indicate possible disease manifestations associated with SARS-CoV-2.

## SARS-CoV-2 cellular network perturbations of disease manifestations

Investigation of the relationships between human host proteins targeted by SARS-CoV-2 and disease susceptibility genes may offer crucial information for identifying COVID-19-associated disease manifestations. We thus inspected the overlap between SARS-CoV-2 host genes/proteins and the susceptibility gene sets implicated in different diseases and biological events (Fig 2I). We found that host genes/proteins targeted by SARS-CoV-2 are significantly enriched in Mendelian disease ($P = 0.002$, Fisher's exact test), orphan disease ($P = 0.044$), and cancer

($P < 0.001$). Mechanistically, SARS-CoV-2 target host genes are significantly enriched in cell cycle genes ($P < 0.001$) and innate immune genes ($P < 0.001$).

Using the 5 SARS-CoV-2 host gene/protein sets, we next tried to identify potential COVID-19 comorbidities. To achieve this, we assembled the disease–protein network of COVID-19 and 6 disease categories (S4 Table). For pan-cancer analysis, the somatic driver genes were retrieved from the Cancer Gene Census Tier 1 gene table [42,43]. The putative somatic driver genes for individual cancer types were identified from The Cancer Genome Atlas projects and were downloaded from a previous study [44]. For autoimmune, pulmonary, neurological, cardiovascular, and metabolic categories, we extracted their associated genes/ proteins from the Human Gene Mutation Database (HGMD) [45,46]. Each gene from HGMD has at least 1 reported mutation associated with the disease. This information on the genes, in addition to the sources, counts, and terms used in HGMD to identify the diseases, can be found in S4 Table. Using the disease–protein network together with the SARS2-PPI (SARS-CoV-2 virus–host interactome) and HCoV-PPI (HCoV–host interactome) sets, we examined the overall connectivity of the disease-associated proteins and the SARS-CoV-2 target host proteins in the human PPI network (Fig 3A; S1 File; S2 Data). To build the global network for the disease comorbidities, we extracted the PPIs from the human interactome for the virus target proteins and disease-associated proteins. Each node indicates a virus target host protein (blue) or a disease-associated protein (green). Some disease-associated proteins can be directly targeted by the viruses, as shown in orange. Edges among these protein nodes indicate PPIs. For SARS-CoV-2, the targets of its individual viral proteins are shown based on the SARS2-PPI dataset. Due to their tendency of having common disease-associated proteins, some disease categories tend to cluster closely, e.g., cancer and neurological. Diseases from other categories, such as autoimmune and pulmonary, are scattered. Most of the virus target proteins are connected with the disease-associated proteins, which suggests shared pathobiological pathways of COVID-19 and these diseases. Various cancer types formed a relatively distant module from the virus targets, compared to other disease categories.

Shown in Fig 3A, the disease genes can interact with the SARS-CoV-2 viral proteins either directly or indirectly in the human protein–protein interactome. For example, among the 4 chronic obstructive pulmonary disease (COPD)–associated proteins shown in the network, TGFB1 is the direct target of HCoV-229E, and all 4 proteins (TGFB1, DEFB1, SNAI1, and ADAM33) interact with at least 1 SARS-CoV-2 target protein. The risk of various cardiovascular diseases was found to be increased in COVID-19 patients, including heart block, coronary artery disease, congestive heart failure, and arrhythmia, which is consistent with clinically reported myocardial injury [47] and cardiac arrest [48]. These observations reveal a common network relationship between COVID-19 and human diseases (Fig 3A). We then performed meta-analysis of 34 COVID-19 clinical studies (S5 Table) to evaluate the pooled risk ratios of 10 comorbidities among 4,973 COVID-19-positive patients (including 2,268 with mild and 731 with severe COVID-19). The random effects model was used to estimate the pooled risk ratio of disease severity. The $tau^2$ and $I^2$ statistics were used to evaluate the heterogeneity among studies (see Materials and Methods—Risk ratio analysis for COVID-19 patients). We found that patients with several disease comorbidities or risk factors—including COPD, cardiovascular disease, stroke, diabetes mellitus, chronic kidney disease, hypertension, cancer, and history of smoking—have significantly higher risks of severe COVID-19 (Fig 3B). The overall pooled risk ratio for patients with COPD was 4.33 (95% CI 2.42–7.74, $P = 0.001$) in 12 low heterogeneous clinical studies ($I^2 = 0.0\%$, $P = 0.6$). The COVID-19 patients with cardiovascular diseases had a risk ratio of 3.87 (95% CI 1.97–7.59, $P = 0.001$), and there was slightly higher heterogeneity across the 13 studies ($I^2 = 57.6\%$, $P = 0.005$). We next turned to quantify

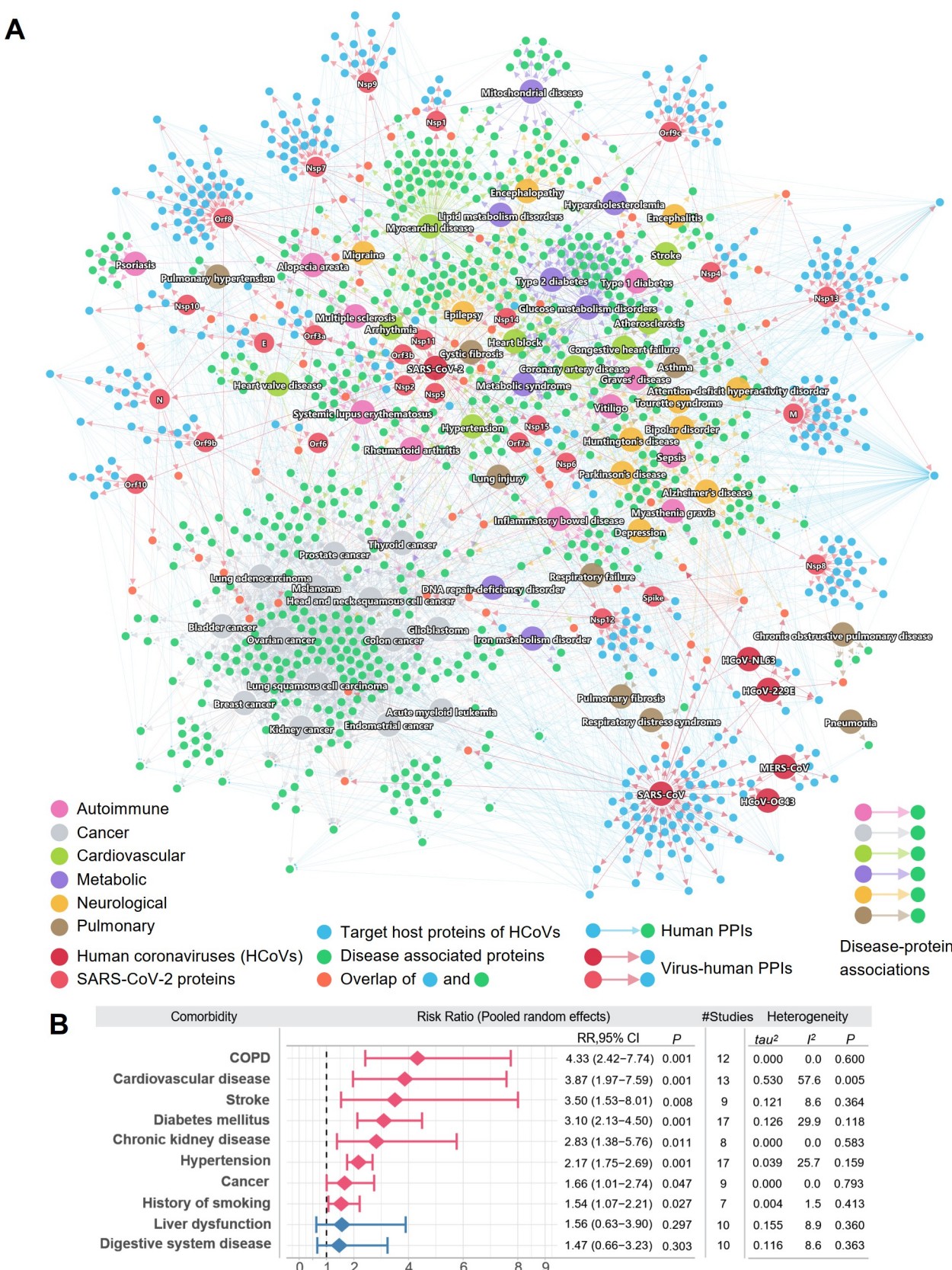

**Fig 3. A global network illustrating disease manifestations associated with human coronavirus.** (A) The target human proteins of SARS-CoV-2 are connected to the disease-associated proteins. Blue links (edges) indicate physical protein–protein interactions. For SARS-CoV-2, its viral proteins are shown by light red nodes. The target human proteins (blue) of the viruses are intricately connected to the disease-associated proteins (green). Human disease nodes are colored by different disease categories: autoimmune, cancer, cardiovascular, metabolic, neurological, and pulmonary. (B) Estimation of the pooled risk ratio using random effects meta-analysis for 10 comorbidities between patients with severe versus non-severe COVID-19. The $tau^2$ and $I^2$ statistics were calculated to quantify the heterogeneity among studies. $I^2 \leq 50\%$ was considered as low heterogeneity among studies, $50\% < I^2 \leq 75\%$ was considered as moderate heterogeneity, and $I^2 > 75\%$ was considered as high heterogeneity. The data underlying this figure can be found in S2 Data. COPD, chronic obstructive pulmonary disease; PPI, protein–protein interaction.

the network-based relationships between COVID-19 and human diseases in the human interactome model using state-of-the-art network proximity analysis [32,33,49].

## Network-based measurement of COVID-19-associated disease manifestations

We systematically evaluated the network-based relationships of the 64 diseases across the 6 categories to COVID-19 (Fig 4A; S3 Data). We used the state-of-the-art network proximity measure to evaluate the connectivity and the closeness of the disease proteins and SARS-CoV-2 host proteins, taking the topology of the human interactome network into consideration. To test the significance of the proximity, $Z$ scores and $P$ values were calculated based on permutation tests (see Materials and Methods—Network proximity measure) and are shown in Fig 4A. We found that each disease–disease pair has a well-defined network-based footprint. If the footprint between the COVID-19 module and another disease module is significantly close (low $Z$ score and $P < 0.05$), the magnitude of the proximity is indicative of their biological relationship: Closer network proximity (Fig 4A) of SARS-CoV-2 host genes/proteins with a disease module indicates higher potential of manifestation between COVID-19 and a specific disease. We first noticed that immunological, pulmonary, and neurological diseases showed significant network proximity to different SARS-CoV-2 maps more frequently than did cancer, cardiovascular, and metabolic diseases. Several diseases have significant network proximities to more than 1 SARS-CoV-2 dataset, most notably inflammatory bowel disease (IBD), attention-deficit/hyperactivity disorder, and stroke, which achieved significant $P$ values for all 5 SARS-CoV-2 protein sets. Pulmonary diseases, including COPD, lung injury, pulmonary fibrosis, and respiratory failure, achieved 4 significant proximities. Some diseases have significant proximities to certain SARS-CoV-2 datasets, indicating associations at certain levels, e.g., asthma (transcriptomic), respiratory distress syndrome (proteomic), and hypertension (HCoV-PPI and PanCoV-PPI).

Network visualization can further show the connections between SARS-CoV-2 and other diseases, for example, respiratory distress syndrome (Fig 4B), sepsis (Fig 4C), and COPD (S7 Fig). Respiratory distress syndrome and sepsis are the 2 main causes of mortality in patients with severe COVID-19 [50,51]. We found that multiple SARS-CoV-2 host proteins are directly connected with the disease-associated proteins (Fig 4B). ABCA3 is a lipid transporter located in the outer membrane of lamellar bodies in AT2 cells; mutations of the *ABCA3* gene can disrupt pulmonary surfactant homeostasis and lead to inherited pulmonary diseases [52]. Another membrane surface protein, the pulmonary-associated surfactant protein C encoded by *SFTPC*, can cause lung injury when misfolded [53]. For sepsis, we noticed several inflammatory and immune-related proteins, such as IRAK1, IRAK3, IKBKB, and STAT3, in the network, suggesting overlap of the inflammatory response activated in COVID-19 and sepsis (Fig 4C). It has been reported that an overzealous production of certain cytokines, such as IL-6, caused by dysregulation of innate immune responses to SARS-CoV-2 infection, can result in a "cytokine storm," better known as cytokine release syndrome (CRS) [54]. The potential

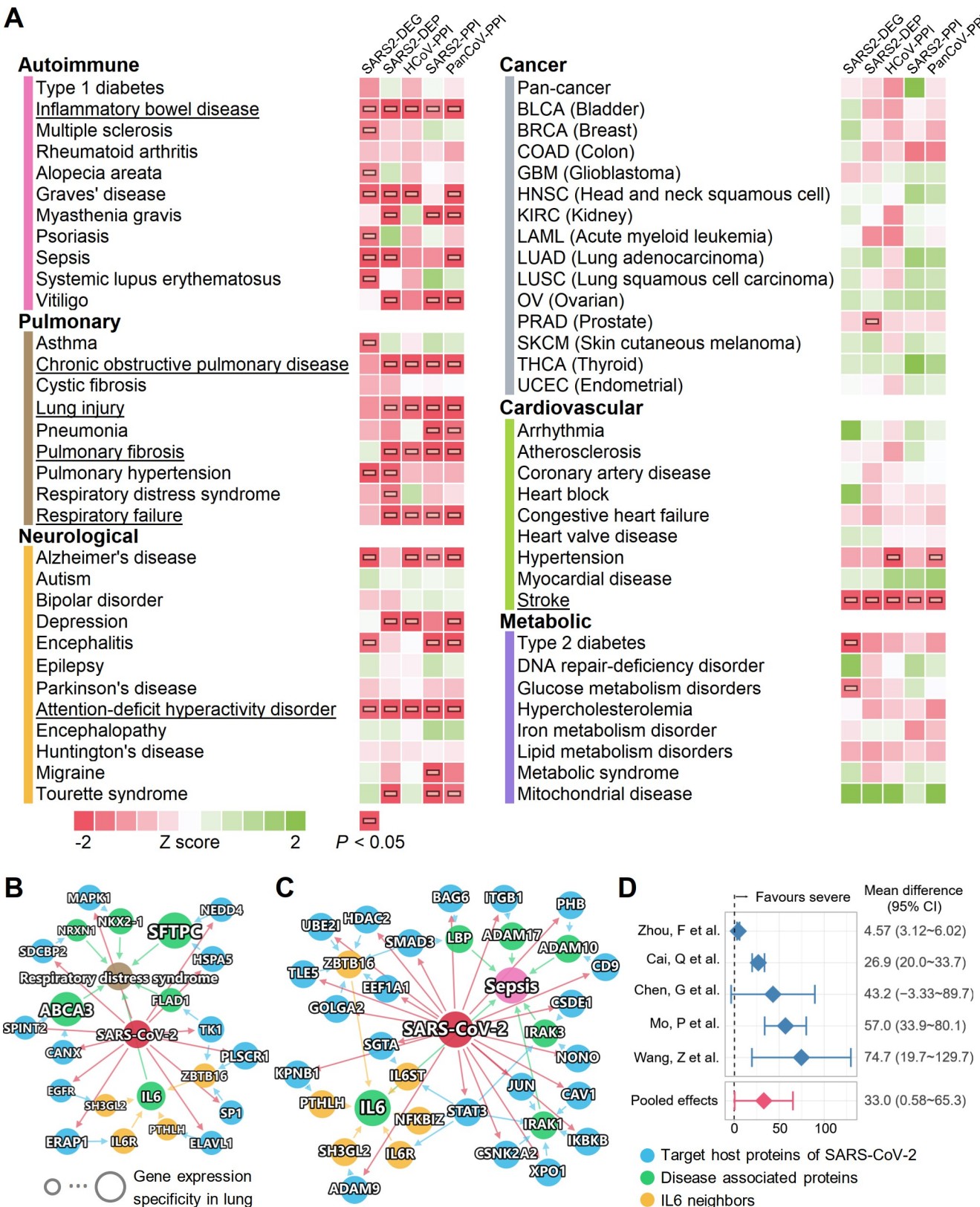

**Fig 4. A landscape of disease manifestations associated with COVID-19 quantified by network proximity measure.** (A) Heatmaps showing the network proximities of COVID-19 with 64 diseases across 6 categories. The network proximities of the disease modules and the 5 SARS-CoV-2 datasets were

evaluated using the "closest" network proximity measure (see Materials and Methods—Network proximity measure). The magnitude of the proximity is indicative of their biological relationship: Closer network proximity of SARS-CoV-2 host genes/proteins and a disease indicates higher potential of manifestation between COVID-19 and the disease. $P < 0.05$ computed by permutation test was considered significant (indicated by horizontal rectangles). Three categories, autoimmune, pulmonary, and neurological, frequently show significant proximities to COVID-19. Inflammatory bowel disease, attention-deficit/hyperactivity disorder, and stroke achieved significance with all 5 SARS-CoV-2 datasets. Underlined diseases are those highlighted in the Results. (B and C) Highlighted subnetworks between SARS-CoV-2 host genes/proteins and the disease-associated proteins of respiratory distress syndrome (B) and sepsis (C). (D) Clinical data analyses showed an association of COVID-19 severity with IL-6 expression levels in patients. Meta-analysis of the random effects model was performed using the mean difference in IL-6 (pg/ml). There was high heterogeneity among these studies ($I^2 = 94\%$, $P < 0.001$). The data underlying this figure can be found in S3 Data.

prognoses of acute respiratory distress syndrome and sepsis using IL-6 expression levels have also been established [55–57]. IL-6 also has a significantly increased expression level in the human bronchial epithelial cells infected with SARS-CoV-2 from the SARS2-DEG dataset [21]. In addition, it is also potentially affected by SARS-CoV-2 through multiple PPIs (Fig 4B and 4C, IL-6 neighbors), such as IL-6R, endophilin A1 (encoded by *SH3GL2*), and parathyroid hormone like hormone (encoded by *PTHLH*). Our random effects meta-analysis of 5 clinical studies of COVID-19 revealed that there was an increase of IL-6 levels in patients with severe COVID-19 compared to those with non-severe COVID-19 (Fig 4D). The mean difference was 33.0 pg/ml (95% CI 0.58–65.3; Fig 4D), with high literature heterogeneity ($I^2 = 94\%$, $P < 0.001$). These results indicate that IL-6 plays a critical role in COVID-19-associated respiratory distress syndrome and sepsis. Due to the importance of IL-6 in SARS-CoV-2 infection, IL-6 antagonists including tocilizumab [58] (NCT04315480) and sarilumab (NCT04327388) are being investigated in clinical trials for treatment of patients with severe COVID-19.

As an illustration of the shared pathobiology and inflammatory pathways of COVID-19 with various disease manifestations (Fig 4), we next turned to focus on 2 inflammation-driven diseases, asthma and IBD.

## Inflammatory molecular profile shared by COVID-19 and asthma

Patients with severe COVID-19 symptoms showed a higher prevalence of dyspnea (S8A Fig; $P < 0.001$). To understand the associations between COVID-19 and respiratory disease (including asthma), we adopted a multimodal analysis utilizing bulk and single-cell transcriptomics data and metabolomics data under the human interactome network model. To be specific, we identified the enzymes in the network that are associated with altered metabolites in COVID-19 patients. Comparing the DEGs from asthma patients and DEGs from COVID-19 patients, we aimed to find the common genes/proteins or interacting proteins in these patient groups. Using network analyses (degree enrichment and eigenvector centrality), we can rank the importance of these genes. Last, we examined their expression in the cell types that are more susceptible to SARS-CoV-2 (i.e., cells expressing more *ACE2* and *TMPRSS2*).

Fig 5A shows the subnetwork of the connections among SARS-CoV-2 target host proteins and asthma-associated proteins. Most of these proteins have enriched connections within the subnetwork (Fig 5B; degree enrichment), i.e., more connections in the subnetwork than in a random network of the same size in the human interactome. To evaluate the potential influence of the nodes in the network, we also computed the eigenvector centrality of these proteins. A higher eigenvector centrality suggests a higher network influence of a specific protein node. Six overlapped proteins (orange) from both groups were identified: NFKBIA, IRAK3, TNC, IL-6, ADRB2, and CD86. A recent study showed that glucose metabolism plays a key role in influenza A–regulated cytokine storm [59]. The plasma metabolome of asthma patients versus healthy controls also suggests activated inflammatory and immune pathways [60]. Therefore, in addition to PPIs, we also integrated metabolomics data generated in a previously

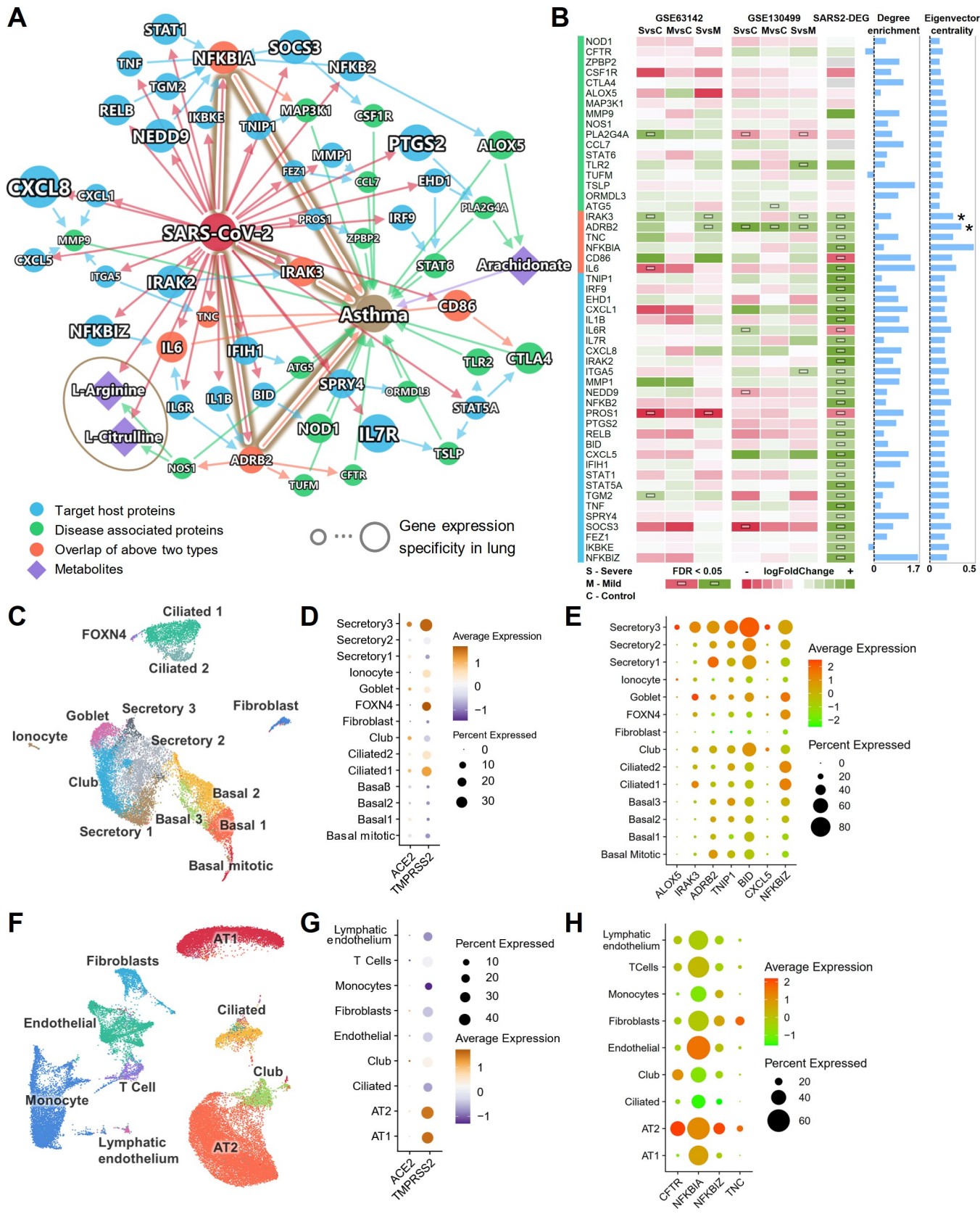

**Fig 5. Shared molecular profile between asthma and COVID-19.** (A) A highlighted subnetwork between the asthma-associated genes, differential metabolites in asthma, and SARS-CoV-2 host genes, under the human interactome network model. (B) A heatmap highlighting differential gene expression analyses for the genes identified in the asthma and COVID-19 subnetwork analysis (A). We performed differential gene expression analysis using 2 existing asthma cohorts (GSE63142 and GSE130499). Blue bars show the node degree enrichment in the subnetwork (A) compared to a random network of the same size (left) and the eigenvector centrality (right). *IRAK3* and *ADRB2* (indicated with asterisks) are associated with asthma and are the targets of SARS-CoV-2. They had significantly elevated expression in asthma patients and SARS-CoV-2-infected human bronchial epithelial cells. (C) UMAP visualization for human bronchial epithelial cells. (D) Cell-type-specific expression levels of *ACE2* and *TMPRSS2* across 14 cell types in human bronchial epithelial cells. (E) Cell-type-specific expression levels of 7 highlighted inflammatory genes (A and B) show elevated expression levels in secretory 3 cells compared to other cell types. (F) UMAP visualization for lung cells. (G) Cell-type-specific expression levels of *ACE2* and *TMPRSS2* across 9 cell types in lung cells. (H) Cell-type-specific expression levels of 4 highlighted inflammatory genes (A and B) show elevated expression levels in alveolar type II (AT2) cells compared to other cell types. The data underlying this figure can be found in S4 Data. Single-cell data were retrieved from https://data.mendeley.com/datasets/7r2cwbw44m/1. See S1 Table for more details of the datasets. FDR, false discovery rate; MvsC, mild versus control; SvsC, severe versus control; SvsM, severe versus mild.

assembled asthma cohort [61]. By matching the enzymes of the differential metabolites and the proteins in the PPI network, we found 3 key metabolites: arachidonate, L-arginine, and L-citrulline. L-arginine and L-citrulline were decreased in the sera of COVID-19 patients [62]. These metabolites were also decreased in asthma patients [61]. Arachidonate, the precursor of a variety of products that regulate inflammatory pathways [63], was found to have an increased level in the inflamed airways of asthma patients [64]. Arachidonate can be converted by 5-lipoxygenase encoded by *ALOX5* to leukotriene, which is released during an asthma attack and is responsible for the bronchoconstriction [65].

We further examined the gene expression data from 2 asthma cohorts from the Severe Asthma Research Program [66,67]. Utilizing 2 bulk RNA-Seq datasets (GSE63142 and GSE130499) of asthma patients compared to healthy controls, we identified that *IRAK3* and *ADRB2* had significantly elevated expression (false discovery rate [FDR] < 0.05) in asthma patients. Both genes also have significantly elevated expression in SARS-CoV-2-infected human bronchial epithelial cells (Fig 5B). IRAK-M, encoded by *IRAK3*, regulates the toll-like receptor/interleukin-1 receptor pathway and NF-κB pathway, and *IRAK3* was identified as an asthma susceptibility gene [68]. *ADRB2* encodes the beta2-adrenergic receptor. The polymorphisms of *ADRB2* (p.Arg16Gly and p. Gln27Glu) increase the risk of asthma occurrence, and p.Gln27Glu is associated with asthma severity [69]. *IRAK3* and *ADRB2* also have high eigenvector centrality scores (top 5 and top 2 among these genes, respectively; Fig 5B, eigenvector centrality). Altogether, altered *IRAK3* and *ADRB2* expression may explain relationships between COVID-19 and asthma, though these findings require experimental and clinical validation in patients with these disorders.

To understand the expressions of the proteins in the asthma–COVID-19 network across different cell types, especially cells that express *ACE2*, we analyzed the single-cell RNA-Seq data from bronchial epithelium (Fig 5C) and lung (Fig 5F) [14]. Consistent with previous studies, *ACE2* and *TMPRSS2* have higher expression in a subtype of the secretory cells (secretory 3 cells) compared to other bronchial epithelial cell types (Figs 5D and S8B–S8F). In the lung, *ACE2* and *TMPRSS2* have relatively higher expression in AT2 cells (Fig 5G and S8G–S8K). We further examined the expression of the genes in the asthma–COVID-19 network (Fig 5A) in these cell types (S9 Fig). Several genes were also more highly expressed in secretory 3 cells (Fig 5E; *ALOX5*, *IRAK3*, *ADRB2*, *TNIP1*, *BID*, *CXCL5*, and *NFKBIZ*) and in AT2 cells (Fig 5H; *CFTR*, *NFKBIA*, *NFKBIZ*, and *TNC*), than in other cell types. *IRAK3* and *ADRB2* are among the 6 overlapped genes, potentially implicating the roles of *IRAK3* and *ADRB2* in COVID-19-associated asthma at the single-cell level as well.

## Immune pathobiology shared by COVID-19 and IBD

It has been shown, using confocal and electron microscopy, that human small intestine is an additional SARS-CoV-2 target organ [70]. Diarrhea is now well described as an occasional

presenting symptom of COVID-19 [71]. Our network proximity analysis showed a significant association of COVID-19 and IBD across all 5 SARS-CoV-2 datasets (Fig 4A). In addition, patients with severe COVID-19 had higher risks of abdominal pain and diarrhea (Figs 6A and S10). To understand these associations at the cellular level, we integrated network analysis and single-cell RNA-Seq analysis using publicly available data [72]. As shown in Fig 6B (S5 Data), although only 1 IBD-associated protein, HEATR3, was found to be the target of the SARS-CoV-2 protein Orf7a, other IBD-associated proteins showed an enriched number of connections to the SARS-CoV-2 target proteins (Fig 6I, degree enrichment).

Using single-cell data from the ileum (distal small bowel) in Crohn disease patients [72], we found that *ACE2* and *TMPRSS2* had low to undetectable expression in the non-epithelial cells (Figs 6C, 6E, and S11A–S11E). However, they showed higher expression levels in the epithelial cells, especially absorptive enterocytes (Figs 6D, 6F and S11F–S11J). We further found that both *ACE2* and *TMPRSS2* had elevated expression levels in inflamed cells compared to uninflamed cells in the absorptive enterocytes (Figs 6G, S11K and S11L). The Pearson correlation coefficient (PCC) of *ACE2* and *TMPRSS2* was also increased in the inflamed cells compared to uninflamed cells (PCC = 0.165 versus PCC = −0.006) in the absorptive enterocytes. In absorptive enterocytes expressing *ACE2*, the expression of *ACE2* was significantly increased (Fig 6H; $P = 0.016$) in the inflamed ileal tissues of Crohn disease patients compared to uninflamed tissues. These observations prompted us to investigate the co-expression of the network genes in the absorptive enterocytes (Fig 6I). Several genes showed elevated co-expression with *ACE2* in inflamed cells, such as *XIAP*, *SMAD3*, *DLG5*, *SLC15A1*, *RAC1*, *STOM*, *RAB18*, and *AKAP8*.

We next turned to highlight 2 potential associations between COVID-19 and IBD. First, SARS-CoV-2 protein Orf7a can directly interact with HEATR3 (top 2 eigenvector centrality; Fig 6I), whose variant was shown to be associated with increased risk of IBD by genome-wide association study [73]. Second, SARS-CoV-2 infection may impact RAC1 (top 4 eigenvector centrality; Fig 6I) signal transduction pathways. RAC proteins play important roles in many inflammatory pathways, and their dysregulation can be pathogenic. Increased *RAC1* expression by single nucleotide polymorphisms promotes an inflammatory response in the colon [74]. Mercaptopurine, an effective treatment for IBD, was found to lower RAC1 expression in IBD patients [75]. Since our results show that *RAC1* and *ACE2* had higher co-expression in inflamed enterocytes (Fig 6I), it is highly possible that these inflamed cells are more susceptible to SARS-CoV-2 infection, and that the infection could lead to an altered RAC1 expression level through PPIs with virus target proteins STOM, HDAC2, POLA2, CIT, and RAP1GDS1 (Fig 6B). Notably *STOM* (top 12 eigenvector centrality) was also more co-expressed with *ACE2* in inflamed cells compared to uninflamed cells (Fig 6I).

## Network-based drug repurposing for COVID-19

Knowledge of the complex interplays between SARS-CoV-2 targets and human diseases indicate possibilities of drug repurposing, as the drugs that target other diseases could potentially target SARS-CoV-2 through the shared functional PPI networks [23]. In addition, drug repurposing efforts may also reveal unrecognized biological connections between originally approved indications/diseases and COVID-19. For example, the aforementioned anti-inflammatory drugs tocilizumab and sarilumab that are now being tested for COVID-19 were originally used for rheumatoid arthritis. Although not significant, our network proximity results show that rheumatoid arthritis has small network proximities (negative *Z* scores) across all 5 SARS-CoV-2 datasets (Fig 4A). Another drug, the thiopurine mercaptopurine, which has been used to treat IBD [76], was one of the top repurposable drugs for COVID-19 proposed in our previous work [30].

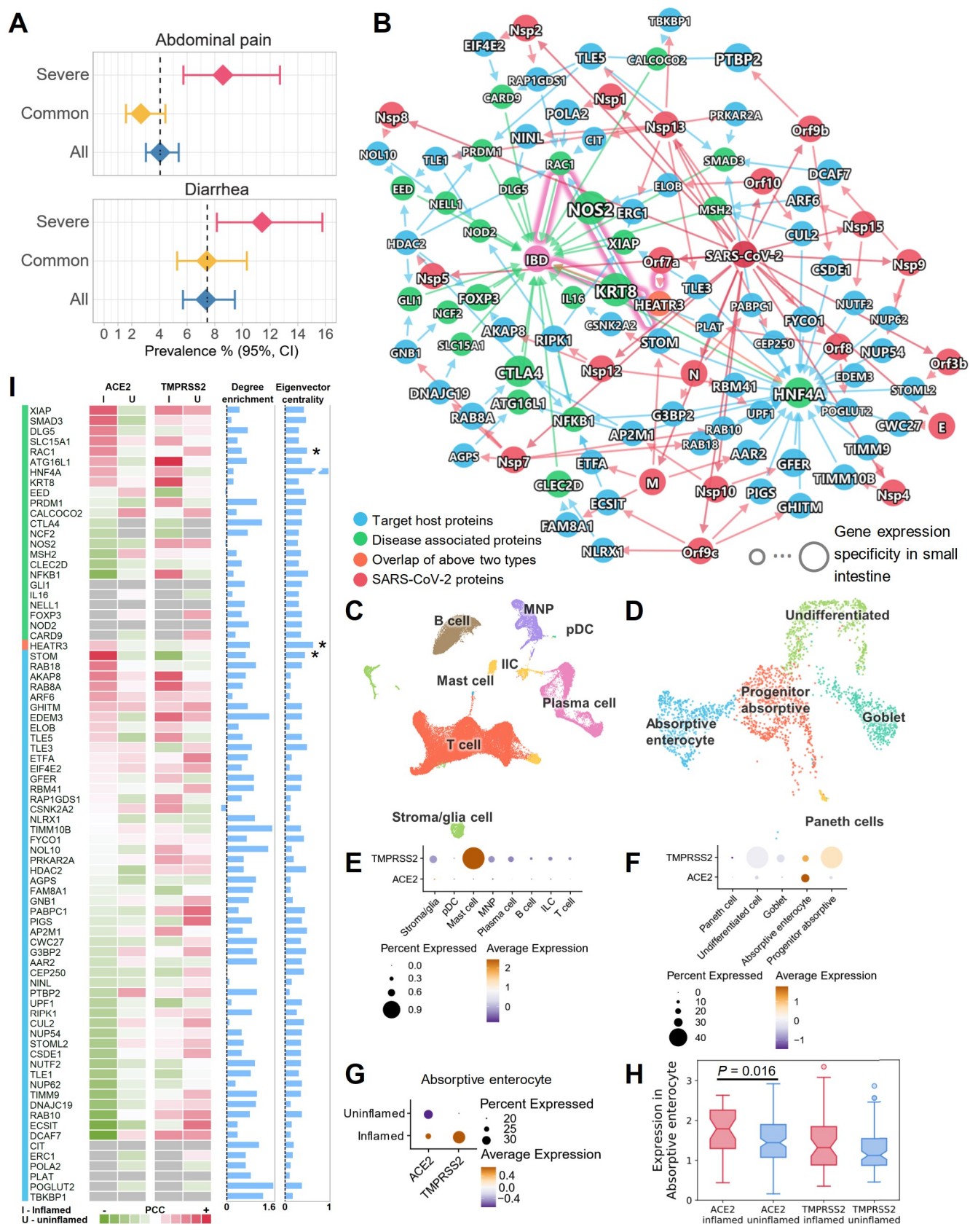

**Fig 6. Inflammatory molecular profile between inflammatory bowel disease (IBD) and COVID-19.** (A) Patients with severe COVID-19 have higher risks of abdominal pain and diarrhea by meta-analysis. (B) A highlighted subnetwork between the IBD-associated genes, the SARS-CoV-2 virus proteins, and virus target proteins under the human interactome network model. (C) UMAP visualization of non-epithelial cells from the ileal tissues of patients with Crohn disease. (D) UMAP visualization of epithelial cells from the ileal tissues of patients with Crohn disease. (E) Cell-type-specific expression of *ACE2* and *TMPRSS2* in non-epithelial cells (C). (F) Cell-type-specific expression of *ACE2* and *TMPRSS2* in epithelial cells (D). (G) The co-expression of *ACE2* and *TMPRSS2* is elevated in absorptive enterocytes of inflamed ileal tissues compared to uninflamed tissues in patients with Crohn disease. (H) Box plot showing the expression of *ACE2* and *TMPRSS2* in absorptive enterocytes expressing *ACE2* and *TMPRSS2*, respectively. (I) Co-expression analysis for the genes in the subnetwork with *ACE2* and *TMPRSS2*. Heatmap shows the Pearson correlation coefficients (PCCs) of *ACE2* and *TMPRSS2* with other genes (labeled in [B]) in the absorptive enterocytes. Blue bars show the degree enrichment of the genes in the subnetwork compared to a random network of the same size (left) and the eigenvector centrality (right). Asterisks indicate that these genes may play important roles in COVID-19-associated IBD. The data underlying this figure can be found in S5 Data. Single-cell data were retrieved from the NCBI GEO database using the accession number GSE134809. See S1 Table for more details of the datasets.

Therefore, we next performed network-based drug repurposing modeling using the existing knowledge of the drug–target network and the global map of the SARS-CoV-2 interactome built in this study. The basis for the network-based drug repurposing methodologies is the observation that for a drug with multiple targets to be effective against a disease, its target proteins should be within or in the immediate vicinity of the corresponding subnetwork of the disease in the human interactome, as we have demonstrated in multiple diseases previously [31,32]. Using our state-of-the-art network proximity framework, we measured the "closest" proximities of nearly 3,000 drugs and the 4 SARS-CoV-2 host gene/protein profiles (SARS2-DEG, SARS2-DEP, HCoV-PPI, and SARS2-PPI; S6 Table). Additionally, we performed gene set enrichment analysis (GSEA) using 5 gene/protein expression datasets: 1 SARS-CoV-2 transcriptomics dataset, 1 SARS-CoV-2 proteomics dataset, 1 MERS-CoV dataset, and 2 SARS-CoV-1 transcriptomics datasets. GSEA was used to evaluate the individual drugs for their potential to reverse the expression at the transcriptome or proteome level altered by the viruses [35].

We next prioritized the drug candidates using subject matter expertise based on a combination of factors: (1) strength of the network-based and bioinformatics-based predictions (a higher network proximity score and significant GSEA score; Fig 7; S6 Data), (2) literature-reported antiviral activities or ongoing clinical trial information, (3) availability of sufficient patient data for meaningful evaluation (exclusion of infrequently used medications) from our COVID-19 registry database, and (4) well-defined antiviral mechanisms of action (such as anti-inflammatory or immune modulators).

In total, we computationally identified 34 drugs that were associated ($Z < -1.5$ and $P < 0.05$, permutation test) with the SARS-CoV-2 datasets (SARS2-DEG, SARS2-DEP, HCoV-PPI, and SARS2-PPI) using the above criteria. These drugs were significantly proximal to 2 or more SARS-CoV-2 host protein sets (Fig 7; S6 Data). We manually curated their reported antiviral profiles. The disease categories that these drugs have been used to treat are also shown in Fig 7. Ten drugs have been used to treat respiratory-related diseases, and the most common categories for these drugs are antibiotic and β2 agonist. The next most common disease category is cardiovascular diseases, for which 7 drugs were predicted. Among the 34 drugs, 3 drugs achieved significant network proximity with all 4 SARS-CoV-2 datasets investigated here. These drugs are (1) the antibiotic drug cefdinir, which is a cephalosporin for the treatment of bacterial infections [77]; (2) the antineoplastic drug toremifene, a selective estrogen receptor modulator that shows striking activities in blocking various viral infections at low micromolar levels, including Ebola virus [78] (50% inhibitive concentration [$IC_{50}$] = approximately 1 μM), MERS-CoV [79] (50% effective concentration [$EC_{50}$] = 12.9 μM), SARS-CoV-1 [80] ($EC_{50}$ = 11.97 μM), and SARS-CoV-2 [81] ($IC_{50}$ = 3.58 μM); and (3) the antihypertensive drug irbesartan, an angiotensin II receptor blocker (ARB) that can inhibit viral entry by inhibiting sodium/bile acid cotransporters [82].

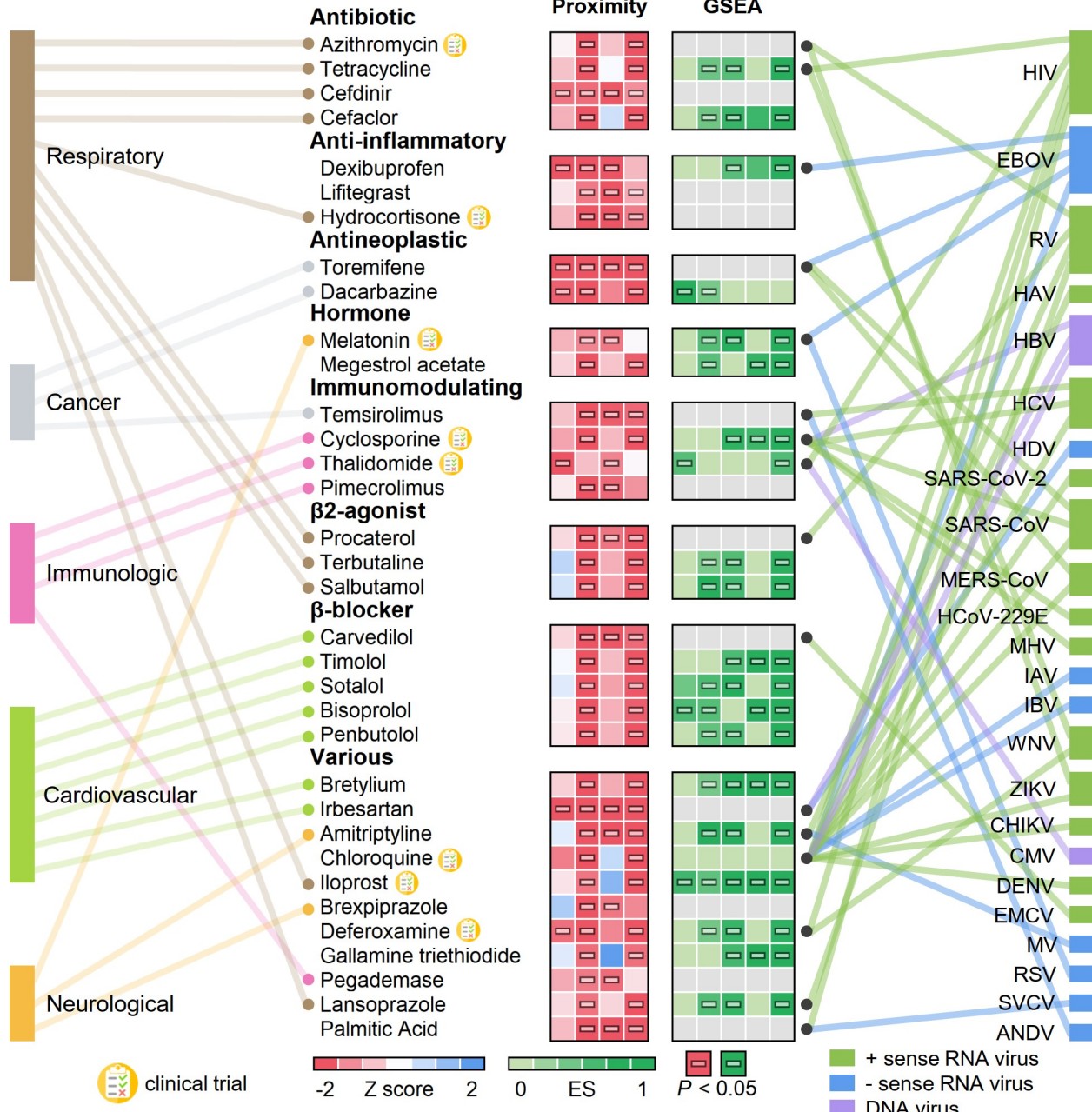

**Fig 7. Network-based prediction of drug repurposing for COVID-19.** Thirty-four drugs from the top predicted list are highlighted, with the disease category they are approved for by the US Food and Drug Administration. We highlight 3 types of evidence: (1) network proximities of drugs' targets across the 4 SARS-CoV-2 datasets (SARS2-DEG, SARS2-DEP, HCoV-PPI, and SARS2-PPI) in the human interactome, (2) gene set enrichment analysis (GSEA) scores across 5 coronavirus transcriptomics and proteomics datasets, and (3) literature-reported antivirus profiles. GSEA scores shown in grey indicate that these drugs cannot be evaluated due to the lack of data. Eight drugs that are currently being or have been tested in COVID-19 clinical trials are highlighted. Horizontal bars in boxes indicate $P < 0.05$. The data underlying this figure can be found in S6 Data. ANDV, Andes virus; CHIKV, chikungunya virus; CMV, cytomegalovirus; DENV, dengue virus; EBOV, Zaire ebolavirus; EMCV, encephalomyocarditis virus; ES, enrichment score; HAV, hepatitis A virus; HBV, hepatitis B virus; HCoV-229E, human coronavirus 229E; HCV, hepatitis C virus; HDV, hepatitis D virus; HIV, human immunodeficiency virus; IAV, influenza A virus; IBV, influenza B virus; MERS-CoV, Middle East respiratory syndrome coronavirus; MHV, mouse hepatitis virus; MV, measles virus; RSV, respiratory syncytial virus; RV, rhinovirus; SARS-CoV, severe acute respiratory syndrome coronavirus; SARS-CoV-2, severe acute respiratory syndrome coronavirus 2; SVCV, spring viremia of carp virus; WNV, West Nile virus; ZIKV, Zika virus.

## Evidence from the COVID-19 registry data that supports the predicted drug repurposing strategies

We next evaluated drug–outcome relationships using a large-scale patient dataset from the Cleveland Clinic COVID-19 patient registry (see Materials and Methods—Patient data validation of the network-identified drugs using a COVID-19 registry). Applying subject matter expertise to the 34 repurposed drugs resulted in identifying melatonin, a physiologic hormone common to many living organisms, and carvedilol, approved for both hypertension and heart failure. A retrospective COVID-19 cohort analysis was conducted to validate the potential prevention effect of melatonin and carvedilol (Fig 8A and 8B). Among a total of 26,779 patients tested for COVID-19 in the Cleveland Clinic Health System in Ohio and Florida, 8,274 patients were diagnosed as SARS-CoV-2 positive confirmed by reverse transcription–polymerase chain reaction (RT-PCR) between March 8 and July 27, 2020 (Table 1).

We found that melatonin usage was associated with a 28% reduced likelihood of a positive laboratory test result for SARS-CoV-2 (odds ratio [OR] = 0.72, 95% CI 0.56–0.91; Fig 8A) after adjusting for age, sex, race, smoking history, and various disease comorbidities (diabetes, hypertension, coronary artery disease, and COPD) using a propensity score (PS) matching method. Angiotensin-converting enzyme inhibitors (ACEIs) and ARBs are 2 common types of drugs for treatment of hypertension. A recent study showed that inpatient use of ACEI/ARB was associated with lower risk of all-cause mortality compared with ACEI/ARB non-use among hospitalized COVID-19 patients with hypertension [83]. Several recent studies also showed that there was no association of ARBs and ACEIs with the risk of SARS-CoV-2 infection [84–86]. We further performed an observational study for 3 cohorts using user active comparator design with ARBs and ACEIs used as comparators and PS adjustment for confounding factors as described in our previous study [32]. We found that melatonin usage was significantly associated with a reduced likelihood of a positive laboratory test result for SARS-CoV-2 compared to use of ARBs (OR = 0.70, 95% CI 0.54–0.92) and ACEIs (OR = 0.69, 95% CI 0.52–0.90) after adjusting for age, sex, race, smoking history, and various disease comorbidities (Fig 8A). Altogether, network-based prediction (Fig 7) and multiple observational analyses (Fig 8A) suggest that melatonin usage offers a potential prevention and treatment strategy for COVID-19; yet, randomized controlled clinical trials are urgently needed to test meaningfully the effect of melatonin for COVID-19.

We found that carvedilol use was significantly associated with a reduced likelihood of a positive laboratory test result for SARS-CoV-2 (OR = 0.74, 95% CI 0.56–0.97) after adjusting for age, sex, race, smoking history, and various disease comorbidities. Yet, carvedilol did not show a significant advantage compared to ARBs (OR = 0.90, 95% CI 0.65–1.25) or ACEIs (OR = 0.73, 95% CI 0.53–1.01).

We therefore tested whether a clinically meaningful effect of melatonin and carvedilol can be observed in different subgroups of patients. To be specific, we generated 5 different subgroups: asthma patients, hypertension patients, diabetes patients, black Americans (African Americans), and white Americans. We found that melatonin was significantly associated with a 52% reduced likelihood of a positive laboratory test result for SARS-CoV-2 in black Americans (OR = 0.48, 95% CI 0.31–0.75; Fig 8C) after adjusting for age, sex, race, smoking, and various disease comorbidities, which is stronger than the association in white Americans (OR = 0.77, 95% CI 0.57–1.04; Fig 8D). In addition, in black Americans, melatonin usage was significantly associated with a reduced likelihood of a positive laboratory test result for SARS-CoV-2 compared to ARB usage (OR = 0.57, 95% CI 0.34–0.96; Fig 8C), while there was no significant difference compared to ACEI usage (OR = 0.65, 95% CI 0.39–1.11; Fig 8C). Yet, melatonin usage was not significantly associated with a reduced likelihood of a positive laboratory

## A. All individuals - melatonin

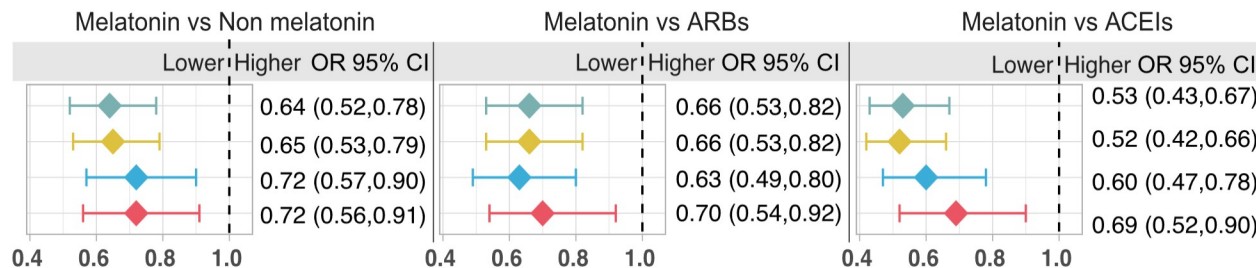

## B. All individuals - carvedilol

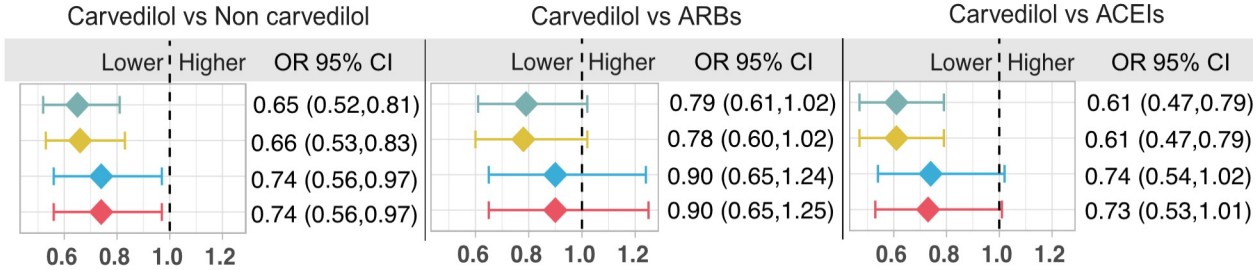

## C. Black Americans - melatonin

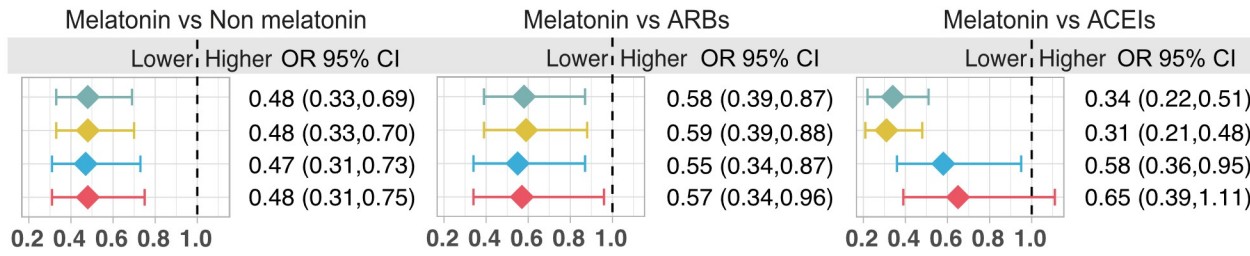

## D. White Americans - melatonin

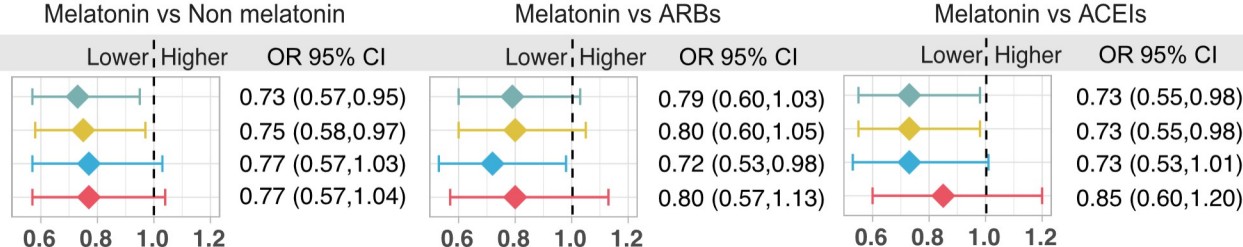

**Fig 8. Patient-based validation of drug repurposing for COVID-19.** Validation for (A) melatonin and (B) carvedilol using the whole COVID-19 registry (all combined population). Validation for melatonin (C) in the black American (African American) subgroup and (D) in the white American subgroup. Patient groups were matched using propensity score matching. Four models were evaluated: (1) model 1 was matched using age, sex, race, and smoking without adjustment for the odds ratio; (2) model 2 was matched using age, sex, race, and smoking, and the odds ratio of COVID-19 was adjusted by age, sex, race, and smoking; (3) model 3 was matched using age, sex, race, smoking, coronary artery disease, diabetes, hypertension, and COPD without adjustment for the odds ratio; and (4) model 4 was matched using age, sex, race, smoking, coronary artery disease, diabetes, hypertension, and COPD, and the odds ratio of COVID-19 was adjusted by age, sex, race, smoking, coronary artery disease, diabetes, hypertension, and COPD. These models were adjusted for different variables using the propensity score matching approach (see Materials and Methods—Patient data validation of the network-identified drugs using a COVID-19 registry). ACEI, angiotensin-converting enzyme inhibitor; ARB, angiotensin II receptor blocker; COPD, chronic obstructive pulmonary disease; OR, odds ratio.

test result for SARS-CoV-2 compared to use of ARBs (OR = 0.80, 95% CI 0.57–1.13; Fig 8D) and ACEIs (OR = 0.85, 95% CI 0.60–1.20; Fig 8D) in white Americans. Among the 3 comorbid

**Table 1. Baseline characteristics of the melatonin and carvedilol use groups from the COVID-19 registry.**

| Use group and characteristic | All individuals | | | SARS-CoV-2-positive patients | | |
|---|---|---|---|---|---|---|
| | Drug not used | Drug used | P value | Drug not used | Drug used | P value |
| **Melatonin** | | | | | | |
| Total patients | 25,724 | 1,055 | | 8,052 | 222 | |
| Age | 48.75 ± 20.58 | 63.42 ± 19.77 | <0.001 | 49.44 ± 20.54 | 67.75 ± 18.92 | <0.001 |
| Sex (female) | 15,274 (59.4) | 549 (52.0) | <0.001 | 4,528 (56.2) | 114 (51.4) | 0.168 |
| Race other | 1,376 (5.3) | 48 (4.5) | 0.287 | 462 (5.7) | 3 (1.4) | 0.008 |
| Race black | 5,848 (22.7) | 273 (25.9) | 0.019 | 2,721 (33.8) | 70 (31.5) | 0.528 |
| Race white | 16,182 (62.9) | 695 (65.9) | 0.054 | 4,290 (53.3) | 142 (64.0) | 0.002 |
| Smoking | 3,050 (13.8) | 165 (16.1) | 0.042 | 603 (9.0) | 19 (8.8) | 1 |
| COPD & emphysema | 1,768 (10.1) | 257 (29.0) | <0.001 | 441 (10.4) | 45 (26.2) | <0.001 |
| Diabetes | 4,681 (25.4) | 417 (44.9) | <0.001 | 1,453 (31.4) | 78 (42.9) | 0.001 |
| Hypertension | 9,810 (49.7) | 764 (76.4) | <0.001 | 3,148 (59.1) | 170 (83.3) | <0.001 |
| Coronary artery disease | 2,699 (15.1) | 330 (36.3) | <0.001 | 744 (17.1) | 59 (34.7) | <0.001 |
| Asthma | 5,075 (27.5) | 265 (29.5) | 0.211 | 1,243 (26.9) | 38 (22.2) | 0.203 |
| ACEIs | 1,873 (7.3) | 208 (19.7) | <0.001 | 607 (7.5) | 48 (21.6) | <0.001 |
| ARBs | 1,371 (5.3) | 148 (14.0) | <0.001 | 447 (5.6) | 21 (9.5) | 0.019 |
| **Carvedilol** | | | | | | |
| Total patients | 25,994 | 785 | | 8,070 | 204 | |
| Age | 48.81 ± 20.67 | 66.38 ± 15.40 | <0.001 | 49.47 ± 20.65 | 68.28 ± 13.08 | <0.001 |
| Sex (female) | 15,485 (59.6) | 338 (43.1) | <0.001 | 4,550 (56.4) | 92 (45.1) | 0.002 |
| Race other | 1,385 (5.3) | 39 (5.0) | 0.717 | 460 (5.7) | 5 (2.5) | 0.066 |
| Race black | 5,841 (22.5) | 280 (35.7) | <0.001 | 2,704 (33.5) | 87 (42.6) | 0.008 |
| Race white | 16,432 (63.2) | 445 (56.7) | <0.001 | 4,328 (53.6) | 104 (51.0) | 0.497 |
| Smoking | 3,080 (13.8) | 135 (17.8) | 0.002 | 604 (9.0) | 18 (9.1) | 1 |
| COPD & emphysema | 1,862 (10.5) | 163 (26.8) | <0.001 | 447 (10.5) | 39 (27.1) | <0.001 |
| Diabetes | 4,656 (25.0) | 442 (63.4) | <0.001 | 1,422 (30.6) | 109 (66.9) | <0.001 |
| Hypertension | 9,837 (49.2) | 737 (95.7) | <0.001 | 3,124 (58.6) | 194 (97.5) | <0.001 |
| Coronary artery disease | 2,615 (14.5) | 414 (60.7) | <0.001 | 704 (16.2) | 99 (61.5) | <0.001 |
| Asthma | 5,184 (27.7) | 156 (25.1) | 0.169 | 1,243 (26.7) | 38 (26.6) | 1 |
| ACEIs | 1,866 (7.2) | 215 (27.4) | <0.001 | 597 (7.4) | 58 (28.4) | <0.001 |
| ARBs | 1,324 (5.1) | 195 (24.8) | <0.001 | 418 (5.2) | 50 (24.5) | <0.001 |

Age is shown as mean ± standard deviation. All other characteristics are shown as number of cases (percentage). Percentages were calculated using the total number of patients with known information for each variable. P values were calculated by 2-sided t test for age and Fisher's exact test for other variables.

ACEI, angiotensin-converting enzyme inhibitor; ARB, angiotensin II receptor blocker; COPD, chronic obstructive pulmonary disease.

disease subgroup analyses, melatonin usage was significantly associated with a reduced risk of SARS-CoV-2 positive test in diabetes patients only (OR = 0.52, 95% CI 0.36–0.75; S12B Fig); there was no significant association for asthma (OR = 0.61, 95% CI 0.36–1.06; S12A Fig) or hypertension (OR = 0.80, 95% CI 0.61–1.05; S12C Fig) patients. Carvedilol usage was not significantly associated with a reduced likelihood of a positive laboratory test result for SARS-CoV-2 among the 5 subgroups after adjusting for age, sex, race, smoking, and various disease comorbidities (S12 and S13 Figs). Thus, further observations using large-scale independent cohorts to test the meaningful effect of carvedilol in reducing risk of COVID-19 are highly needed.

## Discussion

Recent studies indicated that SARS-CoV-2 infection was detected in multiple organs in addition to lungs, including heart, pharynx, liver, kidneys, brain, and intestine [70,87]. SARS-CoV-2 RNA was also found in patient stool [88]. Therefore, investigation of how SARS-CoV-2 associates with other diseases could help reveal and understand its impact on systems and organs in addition to lungs. In this study, we systematically evaluated 64 diseases across 6 categories for their potential manifestations with COVID-19. We started with assembling and characterizing 5 SARS-CoV-2 datasets representing different cellular event levels including transcriptome, proteome, and interactome. Using state-of-the-art network proximity measurement, we identified broad disease manifestations (such as autoimmune, neurological, and pulmonary; Fig 4A) associated with COVID-19. Although the number of genes associated with each disease is different (S4 Table), we did not notice any significant bias in the network proximity $Z$ scores by different number of genes (S14 Fig). Retrospective meta-analyses using the clinical data of 4,973 patients across 34 studies confirmed our network-based findings.

We further investigated the molecular determinants of the association of COVID-19 and the comorbidities by multimodal analyses of large-scale bulk and single-cell transcriptomic profiles, metabolomic data, and the PPI networks. We identified the cell types that have the highest expression levels of *ACE2* and *TMPRSS2*: the lung AT2 cells, secretory bronchial epithelial cells, and absorptive enterocytes in the ileum. We examined the expression of asthma- and IBD-associated genes in their relevant cell types. Combining these findings with the results of differential expression analysis, network analysis, and differential metabolites, we identified several key pathogenic pathways for asthma (including *IRAK3* and *ADRB2*) that can be altered by the viral infection.

For asthma, our network-based findings suggest several possible shared pathobiological pathways associated with COVID-19. First, SARS-CoV-2 infection might alter the expression of several key inflammatory genes: *IRAK3*, which is associated with asthma [68,89,90]; *ADRB2*, which is an essential genetic factor for asthma [69,91]; and *NFKBIA*, which shows critical transcriptional responses in childhood asthma [92]. These genes show high expression in secretory 3 and AT2 cell types, suggesting a higher susceptibility to be impacted by SARS-CoV-2 infection through ACE2. SARS-CoV-2 increased the expression of *IRAK3* and *ADRB2*, which lead to a higher risk of asthma (Fig 5B). Second, decreased levels of L-arginine and L-citrulline were found in SARS-CoV-2-infected patients [62], while it has been shown that higher levels of these metabolites are protective against asthma [61]. L-arginine can be converted to nitric oxide by the nitric oxide synthases, and it was shown that nitric oxide protects against viral infection through multiple potential mechanisms [93].

Our network proximity results show a strong connection between IBD and COVID-19 (Fig 4A). We have also shown that IBD-related pathways can potentially be affected by SARS-CoV-2 infection (Fig 6B). However, we should also note that the meta-analysis does not show a

significant risk ratio of digestive system disease in COVID-19 patients (Fig 3B), while 2 specific symptoms, abdominal pain and diarrhea, showed increased risks in patients with severe COVID-19 (Fig 6A). Digestive system disease covers a broad range of gastrointestinal diseases. Future studies are needed to reveal and validate the associations of COVID-19 and individual gastrointestinal diseases.

Finally, we computationally prioritized nearly 3,000 FDA-approved/investigational drugs for their potential anti-SARS-CoV-2 effects using network proximity measurement and GSEA analysis. A list of 34 repurposable drugs with their reported antiviral profiles are highlighted, among which 8 drugs are currently in ongoing COVID-19 clinical trials (Fig 7). We further explored drug–disease outcome relationships for melatonin using a large-scale COVID-19 patient registry database. We found that among individuals who received testing for SARS-CoV-2, melatonin usage was associated with a 28% and 52% reduced likelihood of a positive laboratory test result for SARS-CoV-2 in the all combined population (Fig 8A) and black Americans (Fig 8C), respectively, after adjusting for age, sex, race, smoking, and various disease comorbidities. Using user active comparator design, we further found that melatonin usage was associated with a reduced likelihood of a positive laboratory test result for SARS-CoV-2 compared to use of ARBs and ACEIs as well. Exogenous melatonin may be of benefit in older patients with COVID-19, given the aging-related reduction of endogenous melatonin and the greater vulnerability of older individuals to mortality from SARS-CoV-2 [94], the latter potentially due to declining immunity, i.e., immunosenescence [95]. Moreover, melatonin suppresses NLRP3 inflammasome activation induced by cigarette smoking and attenuates pulmonary inflammation [96], not only via reduction of NF-κB p65 and tumor necrosis factor-α (TNF-α) expression, but also via increase in anti-inflammatory cytokines such as IL-10 or IL-6, which can also have anti-inflammatory effects [97,98]. Thus, large-scale observational studies and randomized controlled trials are needed to validate the clinical benefit of melatonin for patients with COVID-19. It would be important, however, that the trials be designed with the understanding of the mechanism of the drug to be repurposed. For example, it would be obvious that drugs that decrease viral entry, e.g., part of melatonin's action, would be beneficial in preventing infection or very early in the COVID-19 course, but may be inconsequential when utilized in severe or end-stage infection. Several randomized controlled trials are being performed to test the clinical benefits of melatonin in patients with COVID-19 (ClinicalTrials.gov NCT04409522 and NCT04353128). In addition, the Selective Estrogen Modulation and Melatonin in Early COVID-19 (SENTINEL) trial is underway to test the combination therapy [99] of melatonin with toremifene (an approved selective estrogen receptor modulator [100]) for patients with early and mild COVID-19 (ClinicalTrials.gov NCT04531748).

We acknowledge several potential limitations. First, although we integrated data from multiple sources to build the human interactome and the drug–target network, they are still incomplete. Second, this study relied on the SARS-CoV-2 target host gene/protein datasets, and their quality and literature bias may influence the performance of our network analysis. The genes in these datasets can differ significantly. DEGs identified from transcriptomics profiles can be very different from DEPs from proteomics data by multiple factors, which may influence the results of network proximity analysis as well. These datasets were from different cell types, and the COVID-19-relevant cell types may not be representative. PPIs and DEGs/proteins in different cell lines or tissues may contain false positives as well. Two recent studies suggested that chloroquine or hydroxychloroquine showed ideal antiviral activities in African green monkey kidney cells (VeroE6) but not in a model of reconstituted human airway epithelium [101] or the TMPRSS2-positive lung cell line Calu-3 [102]. These studies showed that cell lines mimicking important aspects of respiratory epithelial cells should be used when analyzing the antiviral activity of drugs targeting host cell functions. In the original SARS2-PPI

dataset based on the VeroE6 cell line, a key PPI for ACE2–spike protein was absent. We posited that combining transcriptomics profiles and proteomics data derived from diverse COVID-19-relevant cell lines or tissues may provide complementary molecular information to overcome the high disease heterogeneities of COVID-19 [103]. In addition, through creating a pan-coronavirus PPI network by combining HCoV-PPI and SARS2-PPI, we aimed to identify broad-spectrum antiviral medications for SARS-CoV-2—and other emerging coronaviruses, if broadly applied—with our network medicine framework.

Third, our method can only apply to diseases with well-characterized genetic information and may not be applicable for diseases that lack such information, such as rare diseases (i.e., cerebral palsy or mental conditions). Potential literature bias of disease-associated genes and the human interactome may also influence our findings. For example, well-studied genes associated with both COVID-19 and other diseases may explain the similarity of COVID-19 with other diseases, while the understudied genes associated with both diseases may not be uncovered. Fourth, the patient data analysis is retrospective, may have selection bias, and is limited for commonly used drugs due to patient data availability. Dose and period of the medications were also missing in our current COVID-19 patient registry database. Although we performed multiple types of PS matching, residual confounding is possible despite high-dimensional covariate adjustment. Carvedilol use did not show a significant advantage compared to use of ARBs (OR = 0.90, 95% CI 0.65–1.25) or ACEIs (OR = 0.73, 95% CI 0.53–1.01) after adjusting for age, sex, race, smoking history, and multiple disease comorbidities (Fig 8B). In addition, carvedilol was not significantly associated with a reduced likelihood of a positive laboratory test result for SARS-CoV-2 among 5 subgroup analyses (S12 and S13 Figs). There are 2 possible explanations: (1) the small number of patients using carvedilol (Table 1) may yield insufficient statistical power for the current observational study or (2) carvedilol may only reduce the likelihood of a positive laboratory test result for SARS-CoV-2 for patients having existing health conditions, such as hypertension. Replication of the associations and causal inference using large-scale independent cohorts may rule out treatment effect heterogeneity and possible confounding further. Finally, although we made the intriguing observation that the use of melatonin was much less prevalent in individuals testing positive for SARS-CoV-2, we recognize that many asymptomatic or minimally symptomatic persons with the virus were not tested and, therefore, their use of melatonin was not evaluable; melatonin use in this latter group might also have been high. It should therefore be noted that all drugs we identified as therapeutic candidates in this study must be validated using experimental assays and randomized clinical trials before they can be recommended for use in patients with COVID-19.

Recent studies have suggested that COVID-19 is a systemic disease that has impacts on multiple cell types, tissues, and organs [104]. Our network methodology could potentially be improved by using tissue-specific genes for the diseases and incorporating directionality of the gene/protein biological effects. We recomputed network proximity using only genes that have a tissue specificity $\geq 0$ in the associated disease: We found overall consistent results (S15 Fig) compared to lung-specific network analysis (Fig 4A). For example, IBD achieved significant proximities across all 5 SARS-CoV-2 gene/protein datasets in Fig 4A. When using genes with tissue specificity $\geq 0$, IBD showed the same results in small intestine, but not in colon (S15 and S16 Figs). Type 2 diabetes showed significant network proximities across all 5 SARS-CoV-2 gene/protein datasets using pancreas-specific genes (S15 Fig), which is consistent with recent clinical observations [105]. Yet, the GTEx database used in this study was from healthy tissues, which may not represent the gene expression state in the disease condition. In addition, the disease-associated genes used this study were based on somatic or germline genetic evidence. For example, we did not observe significant association of COVID-19 with cancer. One possible explanation is that cancer is a more somatic-mutation-driven, chronic disease than

COVID-19, which involves an acute immune response and inflammation-driven heterogeneous disease.

We also experimented with incorporating the directionality of the gene expression using the 2 asthma expression datasets and SARS2-DEG, since the direction of the DEGs are available in those 2 datasets. We found more significant network proximities and smaller $Z$ scores between asthma and COVID-19 when using the up- or down-expressed genes separately (S17 Fig), suggesting that incorporating the directionalities of genes/proteins may improve the performance of network analysis. A previous study has shown that integration of the directionality of the human interactome didn't change the results of network proximity measurement [49]. Owing to the lack of a systematic human interactome with well-documented directionalities, and without comprehensive information about whether a viral protein activates or inhibits a host protein, we didn't test the influence of the directionality of proteins and the human interactomes on our findings in a systematic way. Therefore, future work is needed to explore well-documented directionalities in human interactome network analysis that integrates precise perturbation effects of disease-causing variants and viral proteins.

In conclusion, our study provides a powerful, integrative network medicine strategy for advancing understanding of COVID-19-associated comorbidities and facilitating the identification of drug candidates for COVID-19. This approach also promises to address the translational gap between genomic studies and clinical outcomes, which poses a significant problem when rapid development of effective therapeutic interventions is critical during a pandemic. From a translational perspective, if broadly applied, the network medicine tools applied here could prove helpful in developing effective treatment strategies for other complex human diseases as well, including other emerging infectious diseases.

## Materials and methods

A list of the sources of all the datasets used in this study can be found in S1 Table.

### Building the datasets of SARS-CoV-2 target host genes/proteins

We assembled 4 SARS-CoV-2 datasets of target host genes/proteins: (1) 246 DEGs in human bronchial epithelial cells infected with SARS-CoV-2 [21] (GSE147507), denoted as SARS2-DEG; (2) 293 DEPs in human Caco-2 cells infected with SARS-CoV-2 [22], denoted as SARS2-DEP; (3) 134 strong literature-evidence-based pan-human coronavirus target host proteins from our recent study [30] with 15 newly curated proteins, denoted as HCoV-PPI; and (4) 332 proteins involved in PPIs with 26 SARS-CoV-2 viral proteins identified by affinity purification–mass spectrometry (AP-MS) [8], denoted as SARS2-PPI. Finally, due to the interactome nature of HCoV-PPI and SARS2-PPI, we combined these datasets as the fifth SARS-CoV-2 dataset, which has 460 proteins and is denoted as PanCoV-PPI. Details of these datasets can be found in S2 Table.

**SARS2-DEG.** In the original study, the primary human bronchial epithelial cells were infected with SARS-CoV-2 for 24 hours. The transcriptome profiles of infected (3 replicates) and uninfected cells (3 replicates) were characterized, and the fold change (FC) and FDR for each gene were calculated by DESeq2 and provided in the original study. We applied a cutoff of $|\log_2 FC| > 0.5$ and FDR $< 0.05$ to identify the DEGs.

**SARS2-DEP.** As described in the previous study [22], human Caco-2 cells were infected with SARS-CoV-2 for up to 24 hours. Proteomics assays of the infected and uninfected cells were measured at 24 hours in triplicates. We used the results at 24 hours, as the original study showed most DEPs at 24 hours. The $P$ values were computed using 2-sided unpaired Student $t$ tests with equal variance assumed in this study. We converted the $P$ value to FDR using the

"fdrcorrection" function in the Python package statsmodels v0.11.1 and used a cutoff of FDR < 0.05 to identify the DEPs.

## Collection of 4 additional virus–host gene/protein networks

To characterize the SARS-CoV-2 datasets, we downloaded 4 virus–host gene/protein networks from previous studies for comparison: (1) 900 virus–host interactions identified by gene-trap insertional mutagenesis connecting 10 other viruses and 712 host genes [35]; (2) 2,855 virus–host interactions identified from RNAi connecting 2,443 host genes and 55 pathogens [35]; (3) 579 host proteins mediating translation of 70 innate immune-modulating viORFs [36]; and (4) 1,292 host genes identified by Co-IP+LC/MS that mediate influenza–host interactions [37]. All details for these 4 virus–host gene/protein networks are provided in S3 Table.

## Building the disease gene profiles

We compiled the disease-associated gene sets from various sources. All databases were accessed on March 26, 2020.

**Cancer.** We defined a driver gene as a gene that had significantly enriched driver mutations based on whole-genome or whole-exome sequencing data or reported experimental data from the Cancer Gene Census [42,43] or the original publications from The Cancer Genome Atlas (TCGA, https://portal.gdc.cancer.gov/). The pan-cancer driver genes were retrieved from the Cancer Gene Census [42,43]. Driver genes for individual cancer types were from a previous study [44].

**Mendelian disease genes (MDGs).** A set of 2,272 MDGs were retrieved from the Online Mendelian Inheritance in Man (OMIM) database [106].

**Orphan disease-causing mutant genes (ODMGs).** A set of 2,124 ODMGs were retrieved from a previous study [107].

**Cell cycle genes.** A set of 910 human cell cycle genes were downloaded from a previous study in which they were identified by a genome-wide RNAi screening [108].

**Innate immune genes.** A set of 1,031 human innate immune genes were collected from InnateDB [109].

**Genes associated with autoimmune, pulmonary, neurological, cardiovascular, and metabolic diseases.** The disease-associated genes/proteins were extracted from HGMD [45]. HGMD is a well-documented database, and we downloaded the whole database for data analysis and extraction using well-documented disease ontology terms [46]. We defined a disease-associated gene as a gene that has at least 1 disease-associated mutation in original publications provided in HGMD. The details, including the sources, number of genes, mutations associated with the disease, and terms used to identify diseases in HGMD, are provided in S4 Table.

## Functional enrichment analysis

We performed Kyoto Encyclopedia of Genes and Genomes (KEGG) and Gene Ontology (GO) biological process enrichment analyses to reveal the biological relevance and functional pathways of the 5 SARS-CoV-2 datasets. All functional enrichment analyses were performed using Enrichr [110]. An overview of the virus infection-related pathways and ontology terms shared by 1 or more datasets was generated by searching for significant pathways or terms (FDR < 0.05). The enrichment analysis results for the 5 SARS-CoV-2 gene/protein sets can be found in S1–S5 Figs.

## Selective pressure and evolutionary rate characterization

We calculated the *dN/dS* ratio [111] and the evolutionary rate ratio [112] as described in our previous study [113]. A *dN/dS* ratio below, equal to, or above 1 suggests purifying selection, neutral evolution, or positive Darwinian selection, respectively [114]. The evolutionary rate ratio was computed using the criterion that a ratio > 1 indicates a fast rate and a ratio < 1 indicates a slow rate [112]. The *dN/dS* and evolutionary rate ratios of the genes in the 5 SARS-CoV-2 datasets and 4 additional virus gene/protein sets can be found in S2 and S3 Tables.

## Tissue specificity analysis

The RNA-Seq data (transcripts per million [TPM]) of 33 tissues from the GTEx V8 release (accessed on March 31, 2020; https://www.gtexportal.org/home/) were downloaded. Genes with count per million (CPM) $\geq$ 0.5 in over 90% of samples in a tissue were considered tissue-expressed genes, and otherwise tissue-unexpressed. To quantify the expression specificity of gene *i* in tissue *t*, we calculated the mean expression $E_i$ and the standard deviation $\sigma_i$ of a gene's expression across all considered tissues. The significance of gene expression specificity in a tissue is defined as

$$z_{it} = \frac{E_{it} - E_i}{\sigma_i} \tag{1}$$

## Risk ratio analysis for COVID-19 patients

PubMed, Embase, and medRxiv databases were searched for publications as of April 25, 2020 (S18 Fig). The search was limited to articles in English describing the demographic and clinical features of SARS-CoV-2 cases. We used the search term ("SARS-COV-2" OR "COVID-19" OR "nCoV 19" OR "2019 novel coronavirus" OR "coronavirus disease 2019") AND ("clinical characteristics" OR "clinical outcome" OR "comorbidities"). Only research articles were included; reviews, case reports, comments, editorials, and expert opinions were excluded. Three criteria were used to select studies from a total of 1,054 initial hits: (1) studies that had $\geq$20 COVID-19 patients; (2) studies that grouped the outcomes by degree of severity of COVID-19 (e.g., severe versus non-severe) according to the American Thoracic Society guidelines for community-acquired pneumonia; and (3) studies that were from different institutions. Two criteria were used for exclusion: (1) studies that focused on specific populations (e.g., only death cases, pregnant women, children, or family clusters) and (2) basic molecular biology research. Finally, 34 studies meeting these criteria were used for further analyses.

We performed random effects meta-analysis to estimate the pooled risk ratio with 95% CI of 10 comorbidities for patients with severe versus non-severe COVID-19. The Mantel–Haenszel method was used to estimate the pooled effects of results [115]. The DerSimonian–Laird method was used to estimate the variance among studies [116]. Continuous data such as IL-6 levels were transformed to mean and standard deviation first using Wan's approach based on sample size, median, and interquartile range [117]. Next, we used the inverse variance method to estimate the pooled mean difference and estimated the variance among studies using the DerSimonian–Laird method. We estimated the pooled prevalence of 3 COVID-19 symptoms (abdominal pain, diarrhea, and dyspnea) and 1 comorbidity (COPD) in 3 COVID-19 patient groups (severe, non-severe, and all). A random intercept logistic regression model was used to estimate pooled prevalence, and a maximum-likelihood estimator was used to quantify the heterogeneity of studies [118]. The tau$^2$ and $I^2$ statistics were calculated for the heterogeneity among studies. We considered $I^2 \leq 50\%$ as low heterogeneity among studies, $50\% < I^2 \leq 75\%$ as moderate heterogeneity, and $I^2 > 75\%$ as high heterogeneity. All meta-analyses were conducted using the meta and dmetar packages in the R v3.6.3 platform.

## Building the human protein–protein interactome

A total of 18 bioinformatics and systems biology databases were assembled to build a comprehensive list of human PPIs with 5 types of experimental evidence: (1) protein complexes data identified by a robust AP-MS methodology collected from BioPlex V2.016 [119]; (2) binary PPIs tested by high-throughput yeast-two-hybrid (Y2H) systems from 2 publicly available high-quality Y2H datasets [120,121] and 1 in-house dataset [32]; (3) kinase–substrate interactions identified by literature-derived low-throughput or high-throughput experiments from Kinome NetworkX [122], Human Protein Resource Database (HPRD) [123], PhosphoNetworks [124], PhosphoSitePlus [125], DbPTM 3.0 [126], and Phospho.ELM [127]; (4) signaling networks identified by literature-derived low-throughput experiments from SignaLink 2.0 [128]; and (5) literature-curated PPIs identified by AP-MS, Y2H, literature-derived low-throughput experiments, or protein 3D structures from BioGRID [129], PINA [130], INstruct [131], MINT [132], IntAct [133], and InnateDB [109]. Inferred PPIs based on gene expression data, evolutionary analysis, and metabolic associations were excluded. Genes were mapped to their Entrez ID based on the NCBI database [134]. The official gene symbols were based on GeneCards (https://www.genecards.org/). The final human protein–protein interactome used in this study included 351,444 unique PPIs (edges or links) connecting 17,706 proteins (nodes). Detailed descriptions for building the human protein–protein interactome are provided in our previous studies [31–33,135]. An overview of the human protein–protein interactome can be found in S19 Fig.

## Network proximity measure

We used the "closest" network proximity measure throughout this study. For 2 gene/protein sets $A$ and $B$, their closest distance $d_{AB}$ was calculated as

$$\langle d_{AB} \rangle = \frac{1}{\|A\| + \|B\|} \left( \sum_{a \in A} \min_{b \in B} d(a, b) + \sum_{b \in B} \min_{a \in A} d(a, b) \right) \qquad (2)$$

where $d(a,b)$ is the shortest distance of $a$ and $b$ in the human interactome. To evaluate the significance, we performed a permutation test using randomly selected proteins from the whole interactome that were representative of the 2 protein sets being evaluated in terms of their degree distributions. We then calculated the $Z$ score as

$$Z_{d_{AB}} = \frac{d_{AB} - \bar{d}_r}{\sigma_r} \qquad (3)$$

where $\bar{d}_r$ and $\sigma_r$ were the mean and standard deviation of the permutation test. All network proximity permutation tests in this study were repeated 1,000 times.

## Network-based comorbidity analysis

To reveal potential COVID-19 comorbidities, we computed the network proximity of the disease-associated proteins for each disease and the 5 SARS-CoV-2 datasets. SARS-CoV-2 target proteins with a non-negative tissue specificity in lung were used in the computation. The degree enrichment for protein $i$ in a subnetwork was calculated as

$$e_i = \frac{d_i / n}{D_i / N} \qquad (4)$$

where $d_i$ is the degree of $i$ in the subnetwork, $n$ is number of nodes in the subnetwork, $D_i$ is the

degree in the complete human protein interactome, and $N$ is the total number of nodes in the interactome. The $\log_{10} e_i$ value is reported.

We also computed the eigenvector centrality [136] of the nodes to evaluate their influence in the network topology while also considering the importance of their neighbors. A high eigenvector centrality value suggests that the node is connected to many other nodes with high eigenvector centrality scores as well. The computation was performed using Gephi 0.9.2 (https://gephi.org/).

## Bulk and single-cell RNA-Seq data analysis

A list of the datasets used in this study can be found in S1 Table.

Bulk RNA-Seq datasets for asthma patients were retrieved from the NCBI GEO database (https://www.ncbi.nlm.nih.gov/geo/) using the accession numbers GSE63142 [66] and GSE130499 [67]. Differential expression of 3 comparisons—severe versus control, mild versus control, and severe versus mild—were performed using the GEO2R function (https://www.ncbi.nlm.nih.gov/geo/geo2r/) [137]. In GSE63142 [66], bronchial epithelial cells of 27 control samples, 72 mild asthma samples, and 56 severe asthma samples were obtained by bronchoscopy with endobronchial epithelial brushing. In GSE130499 [67], bronchial epithelial cells of 38 control samples, 72 mild asthma samples, and 44 severe asthma samples were available by bronchoscopy with endobronchial epithelial brushing as well. The differential expression analysis was performed by defining the groups in GEO2R first, then by selecting the 2 groups to compare. Genes with $|\log_2 FC| > 0.5$ and FDR $< 0.05$ were considered significantly differentially expressed.

Single-cell data of normal lung and primary human bronchial epithelial cells were downloaded from https://data.mendeley.com/datasets/7r2cwbw44m/1 [14]. These datasets contain 39,778 lung cells and 17,451 bronchial epithelial cells with cell type annotated. GSE134809 [72] was downloaded from the NCBI GEO database. This dataset contains 67,050 inflamed and uninflamed cells from the ileal samples of 8 patients with Crohn disease. Qualifying cells based on the criteria from the original paper were used for the single-cell analysis. We used the cell type gene markers from a previous study [72] (*CD3D, CD2, CD7, TNFRSF17, MZB1, BANK1, CD79B, CD22, MS4A1, HLA-DRB1, HLA-DQA1, LYZ, IL3RA, IRF7, GZMB, LILRA4, CLEC4C, TPSAB1, CMA1, KIT, PLVAP, VWF, LYVE1, CCL21, COL3A1, COL1A1, ACTA2, GPM6B, S100B*) for the non-epithelial cells. We used markers from Zhang et al. [15] (*DEFA5, REG3A, DEFA6, SOX4, CDCA7, KIAA0101, TOP2A, MKI67, HMGB2, STMN1, SPINK4, ITLN1, REG4, CLCA1, FCGBP, HMGA1, EIF3F, ETHE1, ADH1C, C1QBP, RBP2, APOB, APOC3, APOA1, APOA4*) for the epithelial cells. The expression of these markers in the cells can be found in S20 and S21 Figs. All single-cell data analyses and visualizations were performed with the R package Seurat v3.1.4 [138]. "NormalizeData" was used to normalize the data. "FindIntegrationAnchors" and "IntegrateData" functions were used to integrate cells from different samples. UMAP was used as the dimension reduction method for visualization.

## Building the metabolite–enzyme network

We built a comprehensive metabolite–enzyme network by assembling data from 4 commonly used metabolism databases: KEGG [139], Recon3D [140], the Human Metabolic Atlas (HMA) [141], and the Human Metabolome Database (HMDB) [142]. The metabolite–enzyme network contains 60,822 records of 6,725 reactions among 3,808 metabolites and 3,446 genes. Four types of enzyme functions were included in the network: biosynthesis, degradation, transformation, and transportation.

## Building the drug–target network

To evaluate whether a drug is closely associated with SARS-CoV-2 target proteins in the human interactome, we gathered the drug–target interaction information from several databases: DrugBank database (v4.3) [143], Therapeutic Target Database (TTD) [144], PharmGKB database, ChEMBL (v20) [145], BindingDB [146], and the IUPHAR/BPS Guide to Pharmacology [147]. We included the interactions that have binding affinities $K_i$, $K_d$, $IC_{50}$, or $EC_{50} \leq$ 10 μM and a unique UniProt accession number with "reviewed" status. The details for building the experimentally validated drug–target network can be found in our recent studies [31–33].

## Network-based drug repurposing

We computed the closest network proximity as described before for 2,938 FDA-approved or investigational drugs and the 5 SARS-CoV-2 datasets. For prioritization, we ranked the drugs by their distance to the datasets ($D < 2$, network distance using the closest measure) and $Z$ score ($Z < -1.5$) from the network proximity analysis. The antiviral profiles of the highlighted drugs were manually curated. COVID-19-related clinical trials were retrieved on August 28, 2020.

## Gene set enrichment analysis (GSEA)

The GSEA was conducted as described in our recent work [30] as an additional source of evidence for drug repurposing. Briefly, for each drug and coronavirus target gene set, we computed an enrichment score (ES) to indicate whether the drug can reverse the effect of SARS-CoV-2 at the transcriptome or proteome level. Gene expression profiles for the drugs were retrieved from the Connectivity Map (CMAP) database [148]. Five gene sets were evaluated: (1) the DEGs in human bronchial epithelial cells infected with SARS-CoV-2 [21] (GSE147507); (2) the DEPs in human Caco-2 cells infected with SARS-CoV-2 [22]; (3 and 4) 2 transcriptome datasets of SARS-CoV-1-infected samples from patient's peripheral blood [149] (GSE1739) and Calu-3 cells [150] (GSE33267), respectively; and (5) the DEGs in MERS-CoV-infected Calu-3 cells [151] (GSE122876).

The ES was calculated for up- and down-regulated genes separately first. The overall ES was calculated as

$$ES = \begin{cases} ES_{up} - ES_{down}, & sgn(ES_{up}) \neq sgn(ES_{down}) \\ 0, & else \end{cases} \tag{5}$$

where $ES_{up}$ and $ES_{down}$ were calculated using $a_{up/down}$ and $b_{up/down}$ as

$$a = \max_{1 \leq j \leq s} \left( \frac{j}{s} - \frac{V(j)}{r} \right) \tag{6}$$

$$b = \max_{1 \leq j \leq s} \left( \frac{V(j)}{r} - \frac{j-1}{s} \right) \tag{7}$$

$j = 1, 2, \cdots, s$ were the genes in the gene sets sorted in ascending order by their rank in the drug profiles. $V(j)$ indicates the rank of gene $j$, where $1 \leq V(j) \leq r$, with $r$ being the number of genes from the drug profile. Then, $ES_{up/down}$ was set to $a_{up/down}$ if $a_{up/down} > b_{up/down}$, and was set to $-b_{up/down}$ if $b_{up/down} > a_{up/down}$. A permutation test was performed to evaluate the significance. Drugs were prioritized and selected if $ES > 0$ and $P < 0.05$.

## Patient data validation of the network-identified drugs using a COVID-19 registry

We used institutional-review-board-approved COVID-19 registry data, including 26,779 individuals (8,274 SARS-CoV-2 positive) tested during March 8 to July 27, 2020, from the Cleveland Clinic Health System in Ohio and Florida. The pooled nasopharyngeal and oropharyngeal swab specimens were tested. SARS-CoV-2 positivity was confirmed by reverse transcription–polymerase chain reaction assay in the Cleveland Clinic Robert J. Tomsich Pathology and Laboratory Medicine Institute. All SARS-CoV-2 testing was authorized by the FDA under an Emergency Use Authorization and complied with the guidelines established by the Centers for Disease Control and Prevention. Data include COVID-19 test results, baseline demographic information, medications, and all recorded disease conditions. We conducted a series of retrospective case–control studies with a new user active comparator design to test the drug–outcome relationships for COVID-19. Patients were actively taking the evaluated drugs (carvedilol and melatonin) at the time of testing. Data were extracted from electronic health records (EPIC Systems) and were manually checked by a study team trained on uniform sources for the study variables. We collected and managed all patient data using REDCap electronic data capture tools. The exposures of drugs (including carvedilol and melatonin) were used as recorded in the medication list in the electronic medical records at the time of testing for SARS-CoV-2. A positive laboratory test result for SARS-CoV-2 was used as the primary outcome. PS was used to match patients to reduce various confounding factors. Four models, from less to more stringent in terms of patient matching and OR adjustment, were performed: (1) model 1 was matched using age, sex, race, and smoking without adjustment for the OR; (2) model 2 was matched using age, sex, race, and smoking, and the OR of COVID-19 was adjusted by age, sex, race, and smoking; (3) model 3 was matched using age, sex, race, smoking, coronary artery disease, diabetes, hypertension, and COPD without adjustment for the OR; and (4) model 4 was matched using age, sex, race, smoking, coronary artery disease, diabetes, hypertension, and COPD, and the OR of COVID-19 was adjusted by age, sex, race, smoking, coronary artery disease, diabetes, hypertension, and COPD. All analyses were conducted using the matchit package in the R v3.6.3 platform.

## Statistical analysis and network visualization

Statistical tests were performed with the Python package SciPy v1.3.0 (https://www.scipy.org/). $P < 0.05$ was considered statistically significant throughout this study. Networks were visualized using Gephi 0.9.2 (https://gephi.org/).

## Ethics statement

Procedures follow institutional guidelines for research on the COVID-19 registry database and were approved by the Cleveland Clinic Foundation Institutional Review Board.

## Supporting information

**S1 Data. Fig 2 data.**
(XLSX)

**S2 Data. Fig 3 data.**
(XLSX)

**S3 Data. Fig 4 data.**
(XLSX)

**S4 Data. Fig 5 data.**
(XLSX)

**S5 Data. Fig 6 data.**
(XLSX)

**S6 Data. Fig 7 data.**
(XLSX)

**S7 Data. S1–S5 Figs data.**
(XLSX)

**S8 Data. S6 Fig data.**
(XLSX)

**S9 Data. S7 Fig data.**
(XLSX)

**S10 Data. S14 Fig data.**
(XLSX)

**S11 Data. S15 Fig data.**
(XLSX)

**S1 Fig. Functional enrichment analysis for SARS2-DEG.** In total, 246 differentially expressed genes (DEGs) in human bronchial epithelial cells infected with SARS-CoV-2 were obtained from the NCBI GEO database with the accession number GSE147507, denoted as SARS2-DEG. The data underlying this figure can be found in S7 Data.
(PDF)

**S2 Fig. Functional enrichment analysis for SARS2-DEP.** In total, 293 differentially expressed proteins (DEPs) in human Caco-2 cells infected with SARS-CoV-2 were obtained from Bojkova et al. [22], denoted as SARS2-DEP. The data underlying this figure can be found in S7 Data.
(PDF)

**S3 Fig. Functional enrichment analysis for HCoV-PPI.** This dataset contains 134 strong literature-evidence-based pan-human coronavirus target host proteins from Zhou et al. [30] with 15 newly curated proteins, denoted as HCoV-PPI. The data underlying this figure can be found in S7 Data.
(PDF)

**S4 Fig. Functional enrichment analysis for SARS2-PPI.** This dataset contains 332 proteins involved in protein–protein interactions with 26 SARS-CoV-2 viral proteins identified by affinity purification–mass spectrometry from Gordon et al. [8], denoted as SARS2-PPI. The data underlying this figure can be found in S7 Data.
(PDF)

**S5 Fig. Functional enrichment analysis for PanCoV-PPI.** Due to the interactome nature of HCoV-PPI and SARS2-PPI, we combined these datasets as the fifth SARS-CoV-2 dataset, which has 460 proteins and is denoted as PanCoV-PPI. The data underlying this figure can be found in S7 Data.
(PDF)

**S6 Fig. Characteristics of the 4 SARS-CoV-2 target datasets.** Node degree (blue), *dN/dS* ratio (orange), evolutionary ratio (green), and lung expression specificity (purple) are shown

for each dataset. Grey areas indicate mean ± standard deviation of 100 repeats using randomly selected genes. The data underlying this figure can be found in S8 Data.
(PDF)

**S7 Fig. Chronic obstructive pulmonary disease and COVID-19.** (A) The risk of chronic obstructive pulmonary disease (COPD) is increased in severe COVID-19 patients. (B) Subnetwork shows the proteins potentially involved in the interaction between COPD and COVID-19. The data underlying this figure can be found in S9 Data.
(PDF)

**S8 Fig. Asthma and COVID-19.** (A) The risk of dyspnea is increased in patients with severe COVID-19. (B) UMAP visualization for human bronchial epithelial cells. (C and D) Expression levels of *ACE2* across 14 cell types. (E and F) Expression levels of *TMPRSS2* across 14 cell types. (G) UMAP visualization for lung cells. (H and I) Expression levels of *ACE2* across 9 cell types. (J and K) Expression levels of *TMPRSS2* across 9 cell types. The single-cell data with cell type annotation were retrieved from Lukassen et al. [14], which contains 39,778 lung cells and 17,451 bronchial epithelial cells.
(PDF)

**S9 Fig. The expression of asthma genes and SARS-CoV-2 targets.** The expression levels of the genes from the asthma–COVID-19 subnetwork in bronchial epithelial cells (A) and lung cells (B) are shown.
(PDF)

**S10 Fig. Risk ratios for abdominal pain and diarrhea in COVID-19 patients.** Abdominal pain (A) and diarrhea (B) have increased risks in patients with severe COVID-19.
(PDF)

**S11 Fig. Inflammatory bowel disease and COVID-19.** (A) UMAP visualization of non-epithelial cells from the ileal tissues of patients with Crohn disease. (B and D) The expression of *ACE2* in the non-epithelial cells in (A). (C and E) The expression of *TMPRSS2* in the non-epithelial cells in (A). (F) UMAP visualization of epithelial cells from the ileal tissues of patients with Crohn disease. (G and I) The expression of *ACE2* in the epithelial cells in (F). (H and J) The expression of *TMPRSS2* in the epithelial cells in (F). (K and L) The expression levels of *ACE2* and *TMPRSS2* in inflamed versus uninflamed ileal absorptive enterocytes in Crohn disease patients. The single-cell data were retrieved from Martin et al. [72], which contains 67,050 inflamed and uninflamed cells from the ileal samples of 8 patients with Crohn disease.
(PDF)

**S12 Fig. Patient-based validation of drug repurposing for COVID-19 using 3 different disease subgroups.** The disease subgroups are (A) asthma, (B) diabetes, and (C) hypertension. Four models were evaluated. These models were matched and adjusted using different variables, as shown in the table. The variable that was used to extract each patient subgroup was not used for propensity score matching or odds ratio adjustment. ACEI, angiotensin-converting enzyme inhibitor; ARB, angiotensin II receptor blocker.
(PDF)

**S13 Fig. Comparison of the patient validation results of carvedilol use in black Americans and white Americans.** In black Americans, carvedilol use showed a lowered risk of a positive SARS-CoV-2 test when propensity score was matched with basic variables (age, sex, and smoking).
(PDF)

**S14 Fig. Analysis of the effect of the number of genes associated with diseases on the network proximity Z scores.** Each dot represents a disease (Z score versus number of genes). No significant bias was observed for the number of genes. The maximum $R^2$ is 0.1468 from the SARS2-PPI dataset. The data underlying this figure can be found in S10 Data.
(PDF)

**S15 Fig. Disease manifestations associated with COVID-19 quantified by network proximity measurement using tissue-specific genes for each disease.** The disease-associated genes were filtered by their tissue specificity. Tissues considered are shown after the disease names. Only genes with positive specificity were retained for the network analysis. After filtering, diseases with fewer than 5 genes were removed from the evaluation. The data underlying this figure can be found in S11 Data.
(PDF)

**S16 Fig. The subnetwork between the IBD-associated genes, the SARS-CoV-2 virus proteins, and virus target proteins.** Node sizes show their tissue specificity in colon.
(PDF)

**S17 Fig. Network proximity analysis of asthma and COVID-19 taking into consideration the directionalities of the differential gene expression.** The up- and down-expressed genes in the 2 asthma datasets (GSE63142 and GSE130499, severe versus control) were computed against the up- and down-expressed genes from the SARS2-DEG dataset. Overall, the results show more significant network proximities and smaller Z scores than when the direction is not considered, as in Fig 4.
(PDF)

**S18 Fig. Workflow of clinical study search.** We searched PubMed, Embase, and medRxiv databases for publications as of April 25, 2020, using the search term ("SARS-COV-2" OR "COVID-19" OR "nCoV 19" OR "2019 novel coronavirus" OR "coronavirus disease 2019") AND ("clinical characteristics" OR "clinical outcome" OR "comorbidities"). Only research articles were included. Several criteria were used to filter the initial 1,054 articles to a final sample of 34 studies for meta-analyses.
(PDF)

**S19 Fig. Overview of the human protein interactome.** Cytoscape 3.7.1 was used for the visualization and for generating the statistics. Clustering coefficient (ranges from 0 to 1) measures the extent to which the nodes in the network tend to cluster together. Network centralization (ranges from 0 to 1) measures the extent to which the topology resembles a star. Network density (ranges from 0 to 1) shows how densely the nodes are connected in the network. Network heterogeneity shows the tendency of the network to contain hub nodes.
(PDF)

**S20 Fig. Cell type markers and their expressions in dot plot used to identify the ileal non-epithelial cells.**
(PDF)

**S21 Fig. Cell type markers and their expressions in dot plot used to identify the ileal epithelial cells.**
(PDF)

**S1 File. Network file for the global network of disease manifestations associated with human coronavirus.**
(ZIP)

**S1 Table. Summary of the datasets used in this study.**
(PDF)

**S2 Table. Five SARS-CoV-2 target datasets used in this study.**
(XLSX)

**S3 Table. Additional virus target lists for comparisons with SARS-CoV-2 targets.**
(XLSX)

**S4 Table. Disease-associated genes.**
(XLSX)

**S5 Table. COVID-19 clinical studies used in the meta-analysis.**
(XLSX)

**S6 Table. Network proximity results for 2,938 drugs against the SARS-CoV-2 datasets.**
(XLSX)

## Acknowledgments

We thank all helpful discussions and critical comments regarding this manuscript from the COVID-19 Research Intervention Advisory Committee members at the Cleveland Clinic.

## Author Contributions

**Conceptualization:** Feixiong Cheng.

**Data curation:** Jiayu Shen.

**Formal analysis:** Yadi Zhou, Yuan Hou, Jiayu Shen.

**Funding acquisition:** Feixiong Cheng.

**Investigation:** Yadi Zhou, Yuan Hou, Jiayu Shen, Feixiong Cheng.

**Methodology:** Yadi Zhou, Yuan Hou.

**Resources:** Lara Jehi, Serpil Erzurum, Feixiong Cheng.

**Software:** Yadi Zhou.

**Supervision:** Feixiong Cheng.

**Validation:** Yadi Zhou, Yuan Hou, Reena Mehra, Asha Kallianpur, Daniel A. Culver, Michaela U. Gack, Samar Farha, Joe Zein, Suzy Comhair, Claudio Fiocchi, Thaddeus Stappenbeck, Timothy Chan, Charis Eng, Jae U. Jung, Lara Jehi, Serpil Erzurum, Feixiong Cheng.

**Writing – original draft:** Yadi Zhou, Yuan Hou, Feixiong Cheng.

**Writing – review & editing:** Serpil Erzurum, Feixiong Cheng.

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
