## [Editor Report · Decision Letter 0]

21 Jun 2020

Dear Dr Cheng, 

Thank you for submitting your manuscript entitled "A Network Medicine Approach to Investigation and Population-based Validation of Disease Manifestations and Therapeutics for COVID-19" for consideration as a Preregistered Research Article by PLOS Biology.

IMPORTANT: You have submitted this as a Preregistered Research Article, but actually you have already performed the study, and this is actually a regular Research Article. Please could you change the article type to Research Article when you upload your metadata (see next paragraph).

Please re-submit your manuscript within two working days, i.e. by Jun 23 2020 11:59PM.

Kind regards,

Roli Roberts

Senior Editor

PLOS Biology

---

## [Decision Letter · Decision Letter 1]

30 Jul 2020

Dear Dr Cheng,

Thank you very much for submitting your manuscript "A Network Medicine Approach to Investigation and Population-based Validation of Disease Manifestations and Therapeutics for COVID-19" for consideration as a Research Article at PLOS Biology. Your manuscript has been evaluated by the PLOS Biology editors, an Academic Editor with relevant expertise, and by several independent reviewers.

As you will see in the reviews below, the referees raise a substantial number of concerns with the analyses performed, the reporting and the ultimate relevance of the drug screening results. All of the issues raised are pertinent and should be thoroughly addressed. Given the interest of the topic and approach, we would like to invite you to submit a thoroughly revised manuscript that takes into account all of the reviewers' comments. Please note that we will not be able to make a decision about publication until we have seen the revised manuscript, your response to the reviewers' comments and how they received the revised manuscript. 

We expect to receive your revised manuscript within 2 months. 

**IMPORTANT - SUBMITTING YOUR REVISION**

*Re-submission Checklist*

*Published Peer Review*

*PLOS Data Policy*

*Blot and Gel Data Policy*

Sincerely,

Roli Roberts

Senior Editor,

rroberts@plos.org,

PLOS Biology

REVIEWS:

Reviewer's Responses to Questions

PLOS authors have the option to publish the peer review history of their article (what does this mean?). If published, this will include your full peer review and any attached files.

Reviewer #1: No

Reviewer #2: No

Reviewer #3: No

Reviewer #1: Overall Summary:

Zhou et al developed network-based methodologies to identify SARS-CoV-2 pathogenesis, disease manifestations, and COVID-19 therapies. They incorporated SARSCoV-2 virus-host protein-protein interactions, transcriptomics, and proteomics into the human interactome. Network proximity measure was used to identify the underlying pathogenesis for broad COVID-19-associated manifestations. Multi-modal analyses of single-cell RNA-seq data identified the co-expression pattern of ACE2 and TMPRSS2 in absorptive enterocytes from the inflamed ileal tissues of Crohn's disease patients compared to uninflamed tissues, revealing shared pathobiology by COVID-19 and inflammatory bowel disease. Integrative analyses of metabolomics and transcriptomics (bulk and single-cell) data from asthma patients indicated that COVID-19 shared intermediate inflammatory endophenotypes with asthma. By combing network-based prediction and propensity score matching observation study of 18118 patients from a COVID-19 registry, the authors identified that melatonin was associated with 64% reduced likelihood of a positive lab test for SARS-CoV-2 and can have better efficacy than angiotensin II receptor blockers or angiotensin converting enzyme inhibitors for treating SARS-CoV-2. However, details about how to identify the differential expressed genes, PPI-networks and drug-target network construction, and single-cell RNA-seq analysis are missing; the rationale about the construction of PanCoV-PPI and the network proximity measure need to be clarified. Most of the results are computational discoveries. It may provide valuable insight if the following concerns are addressed.

Major Comments:

1. Details about the RNA-seq data analysis are missing. I am not sure how the authors generated SARS2-DEP. For example, which samples in GSE147507 are used to identify differentially expressed genes? Which samples are used as controls? What kind of software is used to analyze those samples? How to identify differentially expressed genes? The same problem arises elsewhere in the manuscript.

2. In line 194 of page 10, the authors mentioned that "For cancer, the driver 195 genes for pan-cancer and individual cancer types were retrieved from the Cancer Gene 196 Census [34] and a previous study [35]. For autoimmune, pulmonary, neurological, 197 cardiovascular, and metabolic categories, we extracted their associated genes/proteins 198 from the Human Gene Mutation Database" I am not sure how those genes were retrieved from corresponding databases? Did the authors use any keywords for searching?

3. In line 675 of page 31, the authors mentioned that "The final human protein-protein interactome used in this study included 351,444 unique PPIs connecting 17,706 proteins". However, would the authors put those results in their GitHub page for the reader's reference? Several methods adopted are described as "as described in our previous study". Some statistics/overview of the network features/methods, although published somewhere else, should be provided for a better understanding of the proposed methods. For example, how did the authors build human protein-protein interactome? What are active comparator design and PS adjustment? How did the authors obtain the experimentally validated drug-target network? 

4. Source code and supporting data cannot be found from the Github link provided by the authors.

5. In line 138-142, the authors mentioned that "PanCoV-PPI (Fig. 2B) and four other data sets (SARS2-139 DEG, SARS2-DEP, HCoV-PPI, and SARS2-PPI) (S6 Fig) were more likely to be highly connected (high degree or connectivity) in the human PPI network, including several hubs, such as JUN, XPO1, MOV10, NPM1, VCP, and HNRNPA1". According to Fig. 2B and S6, I cannot see anything supporting data to explain that JUN, XPO1, MOV10, NPM1, VCP, and HNRNPA1 are the hubs.

6. The authors performed functional enrichment analyses for five different PPIs, i.e., SARS2-DEG, SARS2-DEP, HCoV-PPI, SARS2-PPI, and PanCoV-PPI, generated from transcriptomic and proteomic data of SARS-CoV-2 as well as literature-based virus-host protein-protein interactions. They found different PPIs differ considerably in terms of enriched pathways, and then claimed "These observations suggest that these different SARS-CoV-2 data sets capture complementary aspects of the biological and cellular states of the viral life cycle and host immunity". However, many factors can cause the complementary effects. For example, 1. These data are derived from different cells or tissues, and not representative; 2. How did the authors perform functional enrichment analysis for SARS2-PPI and HCoV-PPI? 3. The differential expressed genes identified from transcriptome profiles can be very different from proteomic data.

7. In Figs 2C, 2D, and 2G, what is the physical meaning of the ratio of nonsynonymous to synonymous substitutions (dN/dS) and evolutionary rate ratio? 

8. In line 130 of page 7, the authors mentioned that "further compiled four additional virus-host gene/protein networks identified by different methods for comparisons". However, I am not sure how those virus-host gene/protein networks are identified, details regarding how to generate those networks and relevant references are missing.

9. The network proximity measure was used to measure the distance of two genes/proteins in a protein-protein interaction network, however, the importance of the centrality of the nodes/proteins in the PPI network was ignored. 

10. The authors mentioned that "We next performed network-based drug repurposing using the existing knowledge of the drug-target network." and "Using our state-of-the-art network proximity framework, we measured the "closest" proximities of nearly 3,000 drugs". However, details about the drug-target network, the network proximity framework, and the nearly 3,000 drugs are missing. The authors then computationally found 34 drugs that are associated with SARS-CoV-2 data sets, how did the authors rank those 3,000 drugs? how many of those 34 drugs are being tested, or have been tested in clinical trials and have positive effects for COVID-19 patients? The authors validated the efficacy of melatonin, one of those 34 drugs, on COVID-19 patients using their medical records. I am not sure if melatonin was ranked as the top-one among those 34 drugs? Not sure what are the differences between the drug-repositioning method proposed in this paper and in the authors' previous publication (ref 27), which used a similar network-based approach. 

11. I am not sure how cell types in Figs 5C, 5F, 6C, and 6D are annotated? Are there any control samples in the single-cell RNA-seq analysis?

12. In Fig7.B there are four different bars with different colors, what does the PS-matched model 1 and PS-matched model 2 mean? There are two different OR model1s indicated by different colors, however, are these OR model1s independent? Besides, are four different models (variables) in Fig7.B independent? 

Reviewer #2: Review report on Zhou et al PLoS Biology paper: PBIOLOGY-D-20-01653R1 

Comments for the authors:

This paper addresses the network interpretation of higher risk of morbidity and mortality of COVID-19 patients with one or more other common diseases utilizing integrative network analysis of transcriptomics, proteomics, and human interactome. Utilizing bulk and single cell RNA-seq data together with differential metabolite information (only for asthma patients), the authors provided insights on shared pathobiology of COVID-19 patients with asthma and inflammatory bowel diseases. The authors of this paper utilized their earlier developed in-silico drug repurposing approaches on COVID-19 clinical registry database and prioritized existing FDA-approved drugs as potential therapeutic candidates.

Overall authors utilized all possible data sources and network-inference state of the art methods in their integrative analysis. The question remains however, with regard to whether these methods are good enough to yield substantial predictions. Below are the major concerns that need to be carefully further addressed before this paper can be considered for publication:

Major Comments: 

1. One intrinsic limitation of the authors' method is that directionalities are in general not being taken into account in the various networks they built and/or used. For example, it seems that whether a viral protein activates or inhibits a host protein, or whether a gene is upregulated or downregulated in a disease is not being considered, and this can make the interpretation of results difficult or give rise to ambiguities. As an example of this issue, although the authors have identified proximity between the SARS-CoV-2 network and the asthma network with several shared nodes (Fig. 5A), when comparing the differential expression (DE) profiles in asthma to that in SARS-CoV-2 infection (Fig. 5B), there does not seem to be significant concordance in terms of the direction of DE. Notably, IL6 increased in SARS-CoV-2 infection but decreased in asthma. Will the same findings still hold if the directionality is taken into account properly? We think that this is an important issue that should be addressed appropriately.

2. The different coronavirus datasets were from different cell types, for example the SARS2-DEG was based on data in primary bronchial epithelial cells, while the SARS2-DEP was based on Caco-2 colorectal cancer cell line, and the SARS2-PPI was based on HEK273T cell line. Further, the genes associated with each disease should all act in a context-specific manner in the corresponding tissue types where each of the diseases manifest. It's questionable whether the network within one context can still largely hold if transferred to a different context (e.g. tissue type). The authors have implicitly studied the related issue of cell/tissue type-specific expression. If the tissue type-specific expression information is explicitly used to refine the various SARS2 network separately and specifically for each tissue type, will the findings on proximity with disease networks still hold? This potentially serious confounding factor cannot be ignored and needs to be carefully addressed. 

3. In the 'Validating drug-outcome relationships on COVID-19 using patient data' under Results section, authors utilized their earlier developed approaches of network proximity, GSEA analysis and PS-score matching methods to prioritize the drug candidates. The basis for selection of the final drug melatonin seems weak. The authors didn't mention anywhere how many drugs they finally considered in their analysis and with what frequency each of them was used by the patients. It is not clear whether the drugs (including melatonin) are being taken by the patients before or at the time of being tested positive, if before then what is the time interval between the drug consumption and testing, for how long have the patient been taking the drug, and how such information are being used in terms of selecting the samples to include in the analysis, as well as in the analytical model during the analysis. Such details are critical for the interpretation of the results, and an overview should be provided in the main text (Results or Methods) with comprehensive additional details in the Supplementary Materials. Whatever drug list authors provided in Figure 7A, it is very obvious that all drugs were not used by all the patients in similar frequency. Melatonin is a very common drug compared to other drugs listed in that figure. So all these statistical tests are not at all applicable for all these drugs. In other words, this validation is not very useful in the COVID-19 patients' context, where intake of medicines is not homogeneous among patients. 

4. Can the authors prioritize drugs in a more context-specific manner rather than one drug for all? Likewise, can they prioritize drugs in a more similar group of patients, like for asthma patients or for IBD patients or for hypertension patients? 

5. In many places, the authors just provided some numbers without any biological implication. There is no explanation of such variability in numbers or what are the biological consequences of those. For example, in Figure 2, the authors presented the data for PanCoV-PPI which is a combination of HCoV-PPI and SARS2-PPI. It is well known that SARS2-PPI and HCoV-PPI networks are not very similar. In that case, all these numbers are not very useful towards the overall theme of the paper. 

6. The authors utilized the network proximity measure to evaluate the connectivity and closeness of other diseases with COVID-19. It is now well known a variety of underlying health conditions are risk factors for covid-19 patients, including children with rare diseases, like cerebral palsy or mental conditions. In such scenarios, network proximity is not a very useful measure for identifying this high morbidity and mortality risk. What is the authors take on that?

Minor Comments:

We feel that many parts of the writing are inaccurate or confusing and can be improved (examples given below). We recommend the authors to further refine the writing so that it is easier for the readers to understand:

(a) Line 214, the authors write "these diseases can be targeted directly or interact with the 215 targets of SARS-CoV-2 or other HCoVs." We think it is actually meant that "the disease genes can interact with the viral proteins either directly or indirectly via another host protein".

(b) Line 228, authors can explicitly mention which 8 comorbidities they are referring to here.

(c) Line 293, the term "endophenotype", which has a strict definition, may not be appropriate here, the authors may intend to write "molecular profile" or perhaps "molecular program".

(d) Line 297, in multi-modal analysis, there is no clear explanation what authors exactly did here? There is no clear methodology for their multi-modal analysis in the Methods section.

(e) Line 308, "matching the enzymes of the differential metabolites and the proteins in the PPI network", it's not clear whether enzymes for the synthesis, or degradation, or any transformation, or transportation, etc. of the metabolites were considered, and it seems that this is not explained elsewhere either.

(f) Line 318 "Utilizing two bulk RNA-Seq data sets from asthma patients and healthy controls, we identified elevated expression of IRAK3 and ADRB2 in SARS-CoV-2 infected human bronchial epithelial cells." -- this is confusing.

(g) Line 428 "Validating drug-outcome relationships on COVID-19 using patient data", it seems to mean "evidence from the COVID-19 registry data that supports the predicted drug repurposing strategies".

Reviewer #3: The manuscript by Zhou and colleagues is submitted for consideration for publication to PLOS Biology. In this manuscript the authors tried to investigate pathogenesis, clinical manifestations and therapies COVID-19 using network medicine approach on clinical and multi-omics data. The reason for this study is to understand molecular mechanisms of SARS-CoV-2 infection, to compare with other not-infectious diseases and to identify FDA-approved drugs as potential COVID-19 drug candidates through network-medicine findings and clinical data from a large COVID-19 clinical registry database. The paper is well structured and exhaustive in every parts, while the integrated approach, the authors have used, is original and very interesting. Importantly, this theoretical findings might have practical significance via guiding both pharmaceutical and diagnostic research with the prospect to identify potential new biological targets. It can be recommended for publication upon addressing several concerns into some not clear parts.

The main concerns are:

1. in introduction they report numbers of pandemics, but it needs to insert one or more references (e.g. John Hopkins University), while at row 54 they should add other references about network based-approach model based on comparative PPI interactomes with other HCoV and concept of Disease Map applied to COVID19 .

2. in Results they talk about S2 Table, containing "additional virus-host gene / protein networks identified by different methods for comparisons", but it is not clear how they selected these genes, where these data is from and what purpose it would serve .

3. at pag 19 they wrote: "These observations reveal common network relationship between COVID-19 and human diseases". In my opinion, this phrase could result obvious, due to wide previous literature about COVID19 produced up this moment and the pathogenic and molecular similarity with SARS-CoV.

4. at pag 20 they talk about the network-based relationships of the 64 diseases across the 6 categories to COVID-19, shown in Fig 4A, taking into account the proximity, Z scores and P values, as significance test. However this part results hard to understand: firstly, they should report the absolute numbers of how many genes they used for each Z-score test and p-value, because different sample size of genes provide the streghtness of associations. Secondly, they must explain to biological function of genes tested and the pathway involved, because it is very difficult to figure out why they found strong significant network proximity with attention-deficit / hyperactivity disorder and not with other cardiovascular diseases, since vascular damage is one of most featured manifestations in severe COVID19 cases, or asthma. Moreover, for the networks in this figure, it is not clear why they chose sepsis and respiratory distress syndroms as example and it could result misleading.

5. at pag 28 they analyzed data by a large-scale patient data from the Cleveland Clinic COVID-19 patient registry, evaluating the charateristics of melatonin and carvedilol. I noted that in SARS-CoV-2 positive patients there was a wide diversity in sample size between cases and controls (cases are ~ 2% of controls). I understand it was due to availability of COVID19 cases tested, but it should much more report in the limitation section.

---

## [Decision Letter · Decision Letter 2]

13 Oct 2020

Dear Dr Cheng,

Thank you for submitting your revised Research Article entitled "A network medicine approach to investigation and population-based validation of disease manifestations and drug repurposing for COVID-19" for publication in PLOS Biology. I have now obtained advice from the original reviewers and have discussed their comments with the Academic Editor. 

Based on the reviews, we will probably accept this manuscript for publication, assuming that you will modify the manuscript to address the following points:

a) Please address my Data Policy requests (see further down).

b) Please clarify the funding information (it currently appears that you had no specific funding for this project, but this is unclear).

c) Please go very carefully through the text and especially the Abstract, ensuring that you do *not* overstate your claims regarding melatonin or any other treatments herein. This is particularly important given the high-profile disinformation that has occurred during recent months. You must only claim what is supported by the evidence presented. The final clause ("and identifying melatonin for potential prevention and treatment of COVID-19.") seems excessively strong. Also, given that melatonin is an endogenous chemical, the mentions of melatonin (e.g. "melatonin is associated with a reduced likelihood....") presumably should read "melatonin usage" or similar, to avoid the impression that endogenous levels were measured.

d) Please update any assessment of the current state of COVID-19 treatments to ensure that any fast-moving developments are captured.

We expect to receive your revised manuscript within two weeks. Your revisions should address the specific points made by each reviewer. In addition to the remaining revisions and before we will be able to formally accept your manuscript and consider it "in press", we also need to ensure that your article conforms to our guidelines. A member of our team will be in touch shortly with a set of requests. As we can't proceed until these requirements are met, your swift response will help prevent delays to publication.

- a cover letter that should detail your responses to any editorial requests, if applicable

*Copyediting*

*Published Peer Review History*

*Early Version*

Sincerely,

Roli Roberts

Senior Editor,

rroberts@plos.org,

PLOS Biology

DATA POLICY:

Regardless of the method selected, please ensure that you provide the individual numerical values that underlie the summary data displayed in the following figure panels as they are essential for readers to assess your analysis and to reproduce it: All main and supplementary figure panels except Fig 1. NOTE: the numerical data provided should include all replicates AND the way in which the plotted mean and errors were derived (it should not present only the mean/average values).

IMPORTANT: I note that your supplementary files (Tables S1-S7) and Github deposition contain a significant amount of data; however their relationship to the specific Figure panels is unclear; please either supply the data that immediately underlies the Figures, or specify where it can be found; each main and supplementary Figure should have a clear citation to the underlying data (e.g. "The data underlying this Figure can be found in https://github.com/ChengF-Lab/COVID-19_Map"). It may be more appropriate to re-name Tables S2-S7 as Supplementary Data files.

REVIEWERS' COMMENTS:

Reviewer #2:

We thank the authors for addressing our comments. We think the manuscript has now been largely improved and is suitable for publication.

Reviewer #3:

No comments submitted.

---

## [Editor Report · Decision Letter 3]

28 Oct 2020

Dear Dr Cheng,

On behalf of my colleagues and the Academic Editor, Nicole Soranzo, I am pleased to inform you that we will be delighted to publish your Research Article in PLOS Biology. 

PRODUCTION PROCESS

Before publication you will see the copyedited word document (within 5 business days) and a PDF proof shortly after that. The copyeditor will be in touch shortly before sending you the copyedited Word document. We will make some revisions at copyediting stage to conform to our general style, and for clarification. When you receive this version you should check and revise it very carefully, including figures, tables, references, and supporting information, because corrections at the next stage (proofs) will be strictly limited to (1) errors in author names or affiliations, (2) errors of scientific fact that would cause misunderstandings to readers, and (3) printer's (introduced) errors. Please return the copyedited file within 2 business days in order to ensure timely delivery of the PDF proof. 

If you are likely to be away when either this document or the proof is sent, please ensure we have contact information of a second person, as we will need you to respond quickly at each point. Given the disruptions resulting from the ongoing COVID-19 pandemic, there may be delays in the production process. We apologise in advance for any inconvenience caused and will do our best to minimize impact as far as possible.

EARLY VERSION

PRESS 

Kind regards,

Alice Musson

Publishing Editor, 

PLOS Biology

on behalf of

Roland Roberts,

Senior Editor

PLOS Biology